# Receptor-like cytoplasmic kinases of different subfamilies differentially regulate SOBIR1/BAK1-mediated immune responses in *Nicotiana benthamiana*

Wen R. H. Huang [1,8] ✉, Ciska Braam[1], Carola Kretschmer[2], Sergio Landeo Villanueva [1], Huan Liu[1], Filiz Ferik[2], Aranka M. van der Burgh[1,9], Sjef Boeren[3], Jinbin Wu[1,10], Lisha Zhang [4], Thorsten Nürnberger [4], Yulu Wang[5], Michael F. Seidl [6], Edouard Evangelisti [1,11], Johannes Stuttmann[2,7] & Matthieu H. A. J. Joosten [1] ✉

Cell-surface receptors form the front line of plant immunity. The leucine-rich repeat (LRR)-receptor-like kinases SOBIR1 and BAK1 are required for the functionality of the tomato LRR-receptor-like protein Cf-4, which detects the secreted effector Avr4 of the pathogenic fungus *Fulvia fulva*. Here, we show that the kinase domains of SOBIR1 and BAK1 directly phosphorylate each other and that residues Thr522 and Tyr469 of the kinase domain of *Nicotiana benthamiana* SOBIR1 are required for its kinase activity and for interacting with signalling partners, respectively. By knocking out multiple genes belonging to different receptor-like cytoplasmic kinase (RLCK)-VII subfamilies in *N. benthamiana:Cf-4*, we show that members of RLCK-VII-6, −7, and −8 differentially regulate the Avr4/Cf-4-triggered biphasic burst of reactive oxygen species. In addition, members of RLCK-VII-7 play an essential role in resistance against the oomycete pathogen *Phytophthora palmivora*. Our study provides molecular evidence for the specific roles of RLCKs downstream of SOBIR1/BAK1-containing immune complexes.

Plants have evolved a two-layered innate immune system to fend off invading microbes, of which the first layer is mediated by cell-surface receptors[1]. The two largest families of such receptors are receptor-like proteins (RLPs) and receptor-like kinases (RLKs)[2]. Both RLKs and RLPs detect extracellular immunogenic patterns (ExIPs), leading to extracellularly-triggered immunity (ExTI)[1]. ExTI is associated with various downstream immune outputs, including the rapid phosphorylation of downstream receptor-like cytoplasmic kinases (RLCKs), the

[1]Laboratory of Phytopathology, Wageningen University, Droevendaalsesteeg 1, 6708 PB Wageningen, The Netherlands. [2]Institute for Biology, Department of Plant Genetics, Martin Luther University Halle-Wittenberg, 06120 Halle, Germany. [3]Laboratory of Biochemistry, Wageningen University and Research, Wageningen, the Netherlands. [4]Department of Plant Biochemistry, Centre for Plant Molecular Biology (ZMBP), Eberhard Karls University Tübingen, Auf der Morgenstelle 32, D-72076 Tübingen, Germany. [5]Laboratory of Biomanufacturing and Food Engineering, Institute of Food Science and Technology, Chinese Academy of Agricultural Sciences, Beijing 100193, China. [6]Theoretical Biology & Bioinformatics, Department of Biology, Utrecht University, 3584 CH Utrecht, the Netherlands. [7]Aix Marseille University, CEA, CNRS, BIAM, UMR7265, LEMiRE (Microbial Ecology of the Rhizosphere), 13115 Saint-Paul lez Durance, France. [8]Present address: The Sainsbury Laboratory, University of East Anglia, Norwich, United Kingdom. [9]Present address: Teaching and Learning Centre, Wageningen University & Research, Droevendaalsesteeg 4, 6708 PB Wageningen, The Netherlands. [10]Present address: Department of Plant Pathology, Nanjing Agricultural University, Nanjing 210095, China. [11]Present address: Université Côte d'Azur, INRAE UMR 1355, CNRS UMR 7254, Institut Sophia Agrobiotech (ISA), 06903 Sophia Antipolis, France. ✉e-mail: Wen.Huang@tsl.ac.uk; matthieu.joosten@wur.nl

swift production of reactive oxygen species (ROS), the activation of mitogen-activated protein kinase (MAPK) cascades, large-scale transcriptional reprogramming, and in some cases, the induction of programmed cell death, referred to as the hypersensitive response (HR)[3].

RLKs and RLPs share the same overall structure[2]. However, in contrast to RLKs, RLPs lack an intracellular kinase domain for downstream signalling[4]. To date, the most extensively studied plant cell-surface receptors are leucine-rich repeat (LRR)-RLKs and LRR-RLPs[5,6]. Typically, LRR-RLKs form a complex with the regulatory LRR-RLK BRASSINOSTEROID INSENSITIVE 1-ASSOCIATED KINASE1 (BAK1) in a ligand-dependent manner[7,8]. A well-studied example is the Arabidopsis (*Arabidopsis thaliana*, *At*) LRR-RLK FLAGELLIN-SENSING 2 (FLS2), which specifically recognizes bacterial flagellin (or its N-terminal 22-amino acid epitope, flg22), resulting in the rapid trans-phosphorylation of the cytoplasmic kinase domains of BAK1 and FLS2[7–12].

LRR-RLPs constitutively associate with the LRR-RLK SUPPRESSOR OF BIR1-1 (SOBIR1), and require SOBIR1 for their function in plant immunity[13–15]. The tomato (*Solanum lycopersicum*, *Sl*) resistance protein Cf-4 is one of the best-studied LRR-RLPs and specifically recognizes the apoplastic effector Avr4 secreted by the pathogenic intercellular fungus *Fulvia fulva*, formerly known as *Cladosporium fulvum*[16,17]. Cf-4 forms a complex with SOBIR1 in a ligand-independent manner, which is essential for Cf-4-mediated resistance to *F. fulva*[14,15]. Consistent with LRR-RLKs, the LRR-RLP/SOBIR1 complex also requires the recruitment of BAK1 for the initiation of downstream signalling upon ligand perception[18–20].

Protein phosphorylation, which is a swift and reversible biochemical post-translational modification, plays an important role as a versatile molecular switch in various cellular activities, including the initiation of plant immune responses[5,21,22]. According to whether a conserved arginine (Arg/R) is immediately preceding the highly conserved catalytic aspartate (Asp/D) in their catalytic loop, protein kinases can be subdivided into RD and non-RD kinases[23]. Generally, activation of RD kinases requires phosphorylation of one or more residues in their activation segment[23–25]. In contrast, non-RD kinases require a regulatory RD kinase, such as BAK1, to promote their phosphorylation and signalling[26].

Activation of cell-surface receptor complexes subsequently triggers a suite of downstream signalling events. Increasing evidence suggests that RLCKs are the direct downstream cytoplasmic substrates of activated receptor complexes and that they fill the gap between receptors present at the cell surface and downstream signalling components[21,27–30]. Arabidopsis BOTRYTIS-INDUCED KINASE1 (BIK1) is one of the best-characterized RLCKs and is required for triggering ROS production upon the perception of multiple ExIPs, such as flg22, elf18 and chitin[31–33]. BIK1 and some other RLCKs, which are important to relay immune signalling from the cell surface to the intracellular space, all belong to Arabidopsis RLCK class VII[32,34–36]. RLCK-VII is composed of 46 members, and according to their sequence similarity, these members can be further divided into nine subfamilies, termed RLCK-VII-1 to RLCK-VII-9[37]. Interestingly, flg22-, elf18- and chitin-triggered ROS production is significantly reduced in *rlck-vii-5*, −7 and −8 knock-out Arabidopsis plants, whereas in contrast, *rlck-vii-6* knock-out plants exhibit higher flg22-induced ROS accumulation levels when compared to wild-type (WT) plants[37].

The last two decades have witnessed remarkable progress in our understanding of the initiation and regulation of plant innate immunity. Nevertheless, it is largely unknown how SOBIR1 and BAK1 exactly trans-phosphorylate each other to initiate downstream signalling. Additionally, compared to the model plant Arabidopsis, little is known about the function of the various RLCK-VII subfamilies in Solanaceous plants, and it is thus far not known which RLCKs are involved in the LRR-RLP/SOBIR1-triggered immune signalling pathway.

In this work, we perform a site-directed mutagenesis screen, combined with a complementation study, in *Nicotiana benthamiana:Cf-4 sobir1* plants. *Nb*SOBIR1 threonine (Thr/T) 522, as well as its analogous residues in both tomato SOBIR1s, present in the activation segment of the kinase domain of SOBIR1, is found to play an essential role in Avr4/Cf-4-triggered immune signalling. Interestingly, in vitro phosphorylation assays demonstrate that this highly conserved Thr residue is required for SOBIR1(-like) intrinsic kinase activity. Besides, we show that SOBIR1 directly trans-phosphorylates BAK1, whereas, on the other hand, BAK1 directly trans-phosphorylates SOBIR1. These trans-phosphorylation events are proposed to eventually lead to the full activation of both SOBIR1 and BAK1, and their intrinsic kinase activity is required for these trans-phosphorylation events to take place and the initiation of downstream immune signalling. In addition to Thr522, *Nb*SOBIR1 tyrosine (Tyr/Y) 469, as well as its analogous residues in both tomato SOBIR1s, is identified to be essential for Avr4/Cf-4-triggered MAPK activation and the initiation of the HR, but not for ROS production and its intrinsic kinase activity. By knocking out multiple candidate genes belonging to different RLCK-VII subfamilies in *N. benthamiana:Cf-4*, we show that members of RLCK-VII-6, −7 and −8 are differentially required for full Avr4/Cf-4-triggered biphasic ROS production. These members also play an important role in regulating flg22- and chitin-induced ROS accumulation. Additionally, members from RLCK-VII-7 are essential for the Avr4/Cf-4-induced HR and host resistance of *N. benthamiana* to the oomycete pathogen *Phytophthora palmivora*. Importantly, SOBIR1 and BAK1 directly trans-phosphorylate members of RLCK-VII-6, 7 and 8 in vitro. Our study unveils the molecular mechanism underlying the activation and signal transduction of SOBIR1/BAK1-containing immune complexes through RLCKs in *N. benthamiana*.

## Results

### Thr522, present in the activation segment of the kinase domain of *Nb*SOBIR1, is essential for mounting Avr4/Cf-4-triggered immune signalling

The regulatory LRR-RLK SOBIR1 is present throughout the plant kingdom, including *N. benthamiana* (*Nb*), which is a versatile experimental host plant[13–15]. SOBIR1 is an RD kinase, which suggests that likely SOBIR1 first requires phosphorylation of its activation segment to acquire the kinase-active conformation[23,25]. It has been reported that the kinase domain of *At*SOBIR1 auto-phosphorylates at serine (Ser/S), Thr, and Tyr residues in vitro[38], and in the activation segment of SOBIR1 of tomato and *N. benthamiana* such residues are present. To investigate which residue(s) in the kinase domain of SOBIR1 is(are) essential for the activation of the Cf-4/SOBIR1-triggered signalling pathway, we decided to zoom in on the activation segment of *Nb*SOBIR1[14]. Five potential phosphorylation sites are located in this loop of 30 amino acids, including one Ser and four Thr residues (Fig. 1a). Strikingly, these residues are highly conserved in SOBIR1 from various plant species, including tomato (Supplementary Figs. 1 and 2a).

Cf-4 is functional in *N. benthamiana* and agro-infiltration of Avr4 in stable transgenic *N. benthamiana:Cf-4* plants triggers a typical HR[39]. Cf-4 function is completely abolished in *N. benthamiana:Cf-4 sobir1* knock-out lines, whereas complementation through transient expression of functional *NbSOBIR1* restores the Avr4/Cf-4-mediated HR in such knock-out lines[40]. Based on these observations, we carried out site-directed mutagenesis of the activation segment of *Nb*SOBIR1 to substitute individual Ser/Thr residues with alanine (Ala/A) residue, which lacks the phosphorylatable hydroxyl group and thereby cannot be phosphorylated. Subsequently, we performed a complementation study with the five different *Nb*SOBIR1 mutants, taking along the *Nb*SOBIR1 WT and the corresponding D to N (asparagine/Asn) kinase-dead mutant (D482N), as a positive and negative control, respectively[41]. Interestingly, in contrast to *Nb*SOBIR1 WT but similar to *Nb*SOBIR1 D482N, *Nb*SOBIR1 T522A failed to restore the Avr4/Cf-4-

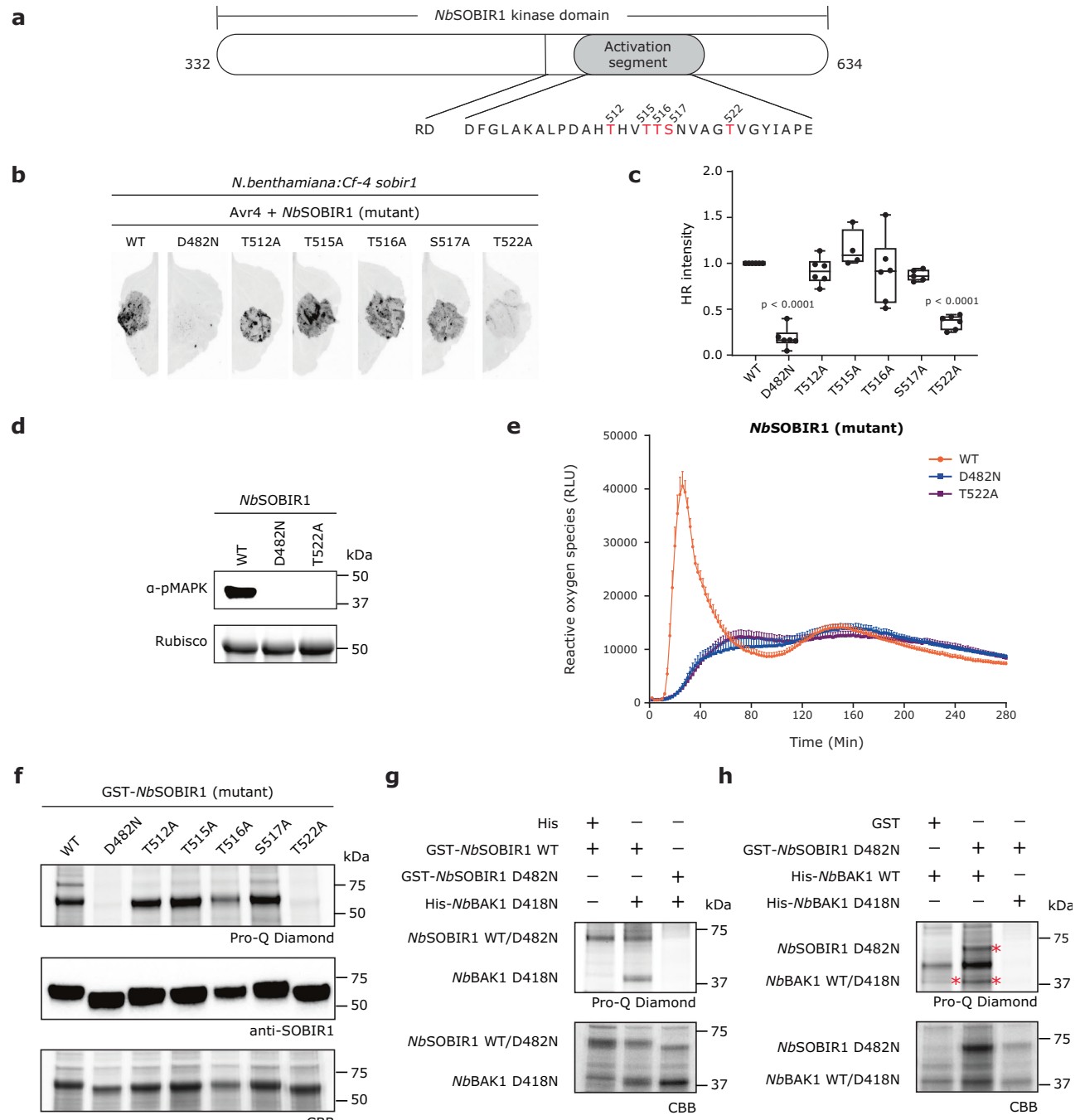

**Fig. 1 | Activation of the SOBIR1/BAK1-containing immune complex by trans-phosphorylation events between SOBIR1 and BAK1. a** Schematic diagram of the kinase domain of *Nb*SOBIR1. The amino acid sequence of the activation segment of *Nb*SOBIR1 is shown below the diagram. Possible phosphorylation sites are denoted in red. **b–e** Complementation with *Nb*SOBIR1 T522A fails to restore Avr4/Cf-4-triggered HR (**b**, **c**), MAPK activation (**d**), and ROS burst (**e**) in *N. benthamiana:Cf-4 sobir1* knock-out plants. The development of HR was imaged (**b**) and quantified (**c**) at 5 dpi. Statistical significance was determined by a one-way ANOVA/Dunnett's multiple comparison test, compared with *Nb*SOBIR1 WT. Dots indicate individual values (centre line, median; error bar, minima and maxima; *n* = 6). Similar to *Nb*SOBIR1 WT, all tested *Nb*SOBIR1 Ser/Thr mutants fully restored the Avr4/Cf-4-triggered ROS production in this complementation study, except for *Nb*SOBIR1 T522A. Only the results for *Nb*SOBIR1 WT, T522A and D482N are shown. ROS

production is expressed as relative light units (RLUs), and the data are represented as mean + SEM (*n* = 12). **f** Thr522 is required for the intrinsic kinase activity of *Nb*SOBIR1. After SDS-PAGE of the *E. coli* lysates, the recombinant GST-*Nb*SOBIR1 cytoplasmic kinase domain and its various mutants were stained with Coomassie brilliant blue (CBB) (bottom panel), whereas their accumulation was detected by western blotting (middle panel), and phosphorylation status was determined by performing a Pro-Q Diamond stain (top panel). **g** *Nb*SOBIR1 WT directly phosphorylates kinase-dead *Nb*BAK1 D418N. **h** *Nb*BAK1 WT directly phosphorylates kinase-dead *Nb*SOBIR1 D482N. Non-fused GST and His tags served as negative controls. Bands with the expected sizes are indicated with an asterisk. All experiments were repeated at least three times with similar results, and representative results are shown. Source data are provided as a Source Data file.

specific HR in *N. benthamiana:Cf-4 sobir1* knock-out plants (Fig. 1b). It is worth noting that this phenotype was not caused by a lack of accumulation of the *Nb*SOBIR1 T522A protein *in planta* (Supplementary Fig. 3). Quantification of the intensity of the HR, which was determined by employing red light imaging[42], showed that the intensity of the HR obtained upon transient co-expression of *NbSOBIR1 TS22A* with *Avr4* was significantly lower than the intensity of the HR obtained when co-expressing *NbSOBIR1 WT* with *Avr4* (Fig. 1c). The four additional mutants of *Nb*SOBIR1 showed a complementation capacity that was similar to *Nb*SOBIR1 WT.

Rapid and transient activation of MAPK cascades is a critical downstream event in the resistance of plants to pathogens[19,43]. To determine whether Avr4/Cf-4-triggered MAPK activation in *N. benthamiana* also requires *Nb*SOBIR1 Thr522, we transiently co-expressed the mutants *NbSOBIR1 T522A* (and *NbSOBIR1 WT* and *D482N* as a positive and negative control, respectively) with *Avr4* in the leaves of an *N. benthamiana:Cf-4 sobir1* knock-out line, and subsequently detected possible MAPK activation by incubating western blots of a total protein extract with p42/p44-erk antibodies. Similar to the negative kinase-dead control but in contrast to the positive WT control, complementation with *Nb*SOBIR1 T522A failed to restore the Avr4/Cf-4-induced MAPK activation in the *sobir1* knock-out line (Fig. 1d).

The swift production of ROS is another hallmark of the plant immune response and apoplastic ROS are mainly produced by nicotinamide adenine dinucleotide phosphate (NADPH) oxidases, such as the RESPIRATORY BURST OXIDASE HOMOLOGUE B (RBOHB) from *N. benthamiana* and tomato, which localizes at the plasma membrane[43-45]. We have shown that the Avr4 protein triggers a biphasic ROS burst in *N. benthamiana:Cf-4*, while this biphasic ROS burst is eliminated in an *N. benthamiana:Cf-4 sobir1* knock-out line[40]. To examine whether *Nb*SOBIR1 activation segment phosphorylation is crucial for mediating the Avr4-triggered ROS burst, we transiently expressed each *Nb*SOBIR1 mutant in the leaves of *N. benthamiana:Cf-4 sobir1* plants, after which the ROS production of discs taken from these leaves was monitored upon adding the Avr4 protein. Intriguingly, in contrast to the other four *Nb*SOBIR1 mutants and the positive control (*Nb*SOBIR1 WT), complementation with neither *Nb*SOBIR1 T522A nor the negative control (*Nb*SOBIR1 D482N) restored the Avr4-triggered ROS burst in *N. benthamiana:Cf-4 sobir1* knock-out line (Fig. 1e).

Consistently, the analogous residues of *Nb*SOBIR1 Thr522 in tomato SOBIR1, which are *Sl*SOBIR1 Thr513 and *Sl*SOBIR1-like Thr526, gave the same phenotype after complementation with their Thr-to-Ala mutants in *N. benthamiana:Cf-4 sobir1* knock-out plants (Supplementary Fig. 2b–i). These residues are overall highly conserved in SOBIR1 from various plant species (Fig. 1a and Supplementary Fig. 2a), which suggests that this specific Thr residue might be crucial for the functionality of SOBIR1 in all plant species.

## SOBIR1 and BAK1 trans-phosphorylate each other in vitro

Thr522 is located at the activation segment of the kinase domain of *Nb*SOBIR1 and plays an important role in Cf-4/SOBIR1-initiated plant immunity. That led us to hypothesize this residue is crucial for the intrinsic kinase activity of SOBIR1, and thereby for its auto-phosphorylation. This SOBIR1 auto-phosphorylation represents step 1 in the model that was proposed by van der Burgh et al.[41], in which SOBIR1 and BAK1 act together in immune signalling. To test whether this is the case, we employed in vitro phosphorylation assays. The N-terminally GST-tagged cytoplasmic kinase domain from *Nb*SOBIR1, as well as this domain from various *Nb*SOBIR1 mutants, were expressed in *Escherichia coli*, which was followed by SDS-PAGE of the *E. coli* lysates, Coomassie brilliant blue (CBB) staining (for determining the accumulation levels of the various mutant kinase domains) and Pro-Q staining (to determine the phosphorylation state of the different

kinase domains)[46]. Successful production of the various *Nb*SOBIR1 kinase domains was confirmed by western blotting, using SOBIR1 antibodies (Fig. 1f). *Nb*SOBIR1 WT exhibited strong auto-phosphorylation activity, similar to its homologue in Arabidopsis[38]. Interestingly, when compared to *Nb*SOBIR1 WT, the auto-phosphorylation activity of *Nb*SOBIR1 T522A was completely abolished, similar to that of the kinase-dead D482N mutant, suggesting that indeed *Nb*SOBIR1 Thr522 is essential for the intrinsic kinase activity, and thereby the auto-phosphorylation, of *Nb*SOBIR1 (Fig. 1f). In line with this observation for *Nb*SOBIR1 T522A, for *Sl*SOBIR1 T513A and *Sl*SOBIR1-like T526A also a loss of their intrinsic kinase activity was observed (Supplementary Fig. 4a, b). Altogether, these results demonstrate that this specific Thr residue is essential for SOBIR1 intrinsic kinase activity.

To elucidate whether SOBIR1 can directly phosphorylate BAK1, corresponding to the proposed SOBIR1 to BAK1 trans-phosphorylation step 2 in the model of van der Burgh et al.[41], we performed an in vitro phosphorylation assay. The GST-tagged *Nb*SOBIR1 WT cytoplasmic kinase domain, or its corresponding kinase-dead mutant, was co-expressed with the His-tagged cytoplasmic kinase domain of the *Nb*BAK1 kinase-dead mutant. The result showed that, first of all, the kinase-dead mutant of *Nb*BAK1, *Nb*BAK1 D418N, of which the conserved "RD" motif in the catalytic loop is changed into "RN", did not have intrinsic auto-phosphorylation activity, as a Pro-Q stain was negative for this mutant when combined with *Nb*SOBIR1 D482N (Fig. 1g). However, this mutant was properly phosphorylated by kinase-active *Nb*SOBIR1 WT, as visualized by a positive Pro-Q stain for *Nb*BAK1 D418N (Fig. 1g). Similarly, the tomato homologue of *Nb*BAK1, *Sl*BAK1, was also directly phosphorylated by *Sl*SOBIR1 WT, as well as by *Sl*SOBIR1-like WT in vitro, as in both cases *Sl*BAK1 D418N was properly phosphorylated. Also here, this trans-phosphorylation fully depended on the intrinsic kinase activity of *Sl*SOBIR1 WT and *Sl*SOBIR1-like WT, as their corresponding "RN" kinase-dead mutants did not phosphorylate *Sl*BAK1 D418N (Supplementary Fig. 4c, d).

We next sought to determine whether BAK1 WT can directly phosphorylate SOBIR1, an event that corresponds to the proposed BAK1 to SOBIR1 trans-phosphorylation step 3 in the model of van der Burgh et al.[41]. Earlier, we already showed that the strong auto-phosphorylation activity of *E. coli*-produced *Nb*SOBIR1 was eliminated in its "RN" kinase-dead mutant (Fig. 1f). Therefore, we performed an additional in vitro phosphorylation assay by co-expressing *Nb*SOBIR1 D482N with either *Nb*BAK1 WT or its kinase-dead mutant. Importantly, *Nb*SOBIR1 D482N was phosphorylated when co-expressed with *Nb*BAK1 WT, but not when co-expressed with *Nb*BAK1 D418N, which demonstrates that indeed *Nb*SOBIR1 can be trans-phosphorylated by *Nb*BAK1 (Fig. 1h). Consistently, *Sl*SOBIR1 D473N and *Sl*SOBIR1-like D486N were also directly phosphorylated by *Sl*BAK1 WT (Supplementary Fig. 4e, f). Notably, trans-phosphorylation of SOBIR1 by BAK1 also required intrinsic kinase activity of BAK1, as phosphorylation of kinase-dead SOBIR1 did not take place when co-expressed with the BAK1 kinase-dead mutant (Fig. 1h, and Supplementary Fig. 4e, f).

## *Nb*SOBIR1 Tyr469 is essential for mounting the Avr4/Cf-4-triggered HR and MAPK activation, but is not required for ROS production and its intrinsic kinase activity

To determine whether particular Tyr residues that are present in the kinase domain of *Nb*SOBIR1 are specifically required for the Avr4/Cf-4-triggered HR, all eight Tyr residues present in the kinase domain of *Nb*SOBIR1 were selected to be studied (Fig. 2a). Strikingly, most of these Tyr residues are highly conserved in SOBIR1 from many different plant species (Supplementary Figs. 5 and 6). We conducted a site-directed mutagenesis involving the substitution of each of the eight Tyr residues of *Nb*SOBIR1 by phenylalanine (Phe/F), which lacks the phosphorylatable hydroxyl group at the aromatic ring. Each *Nb*SOBIR1

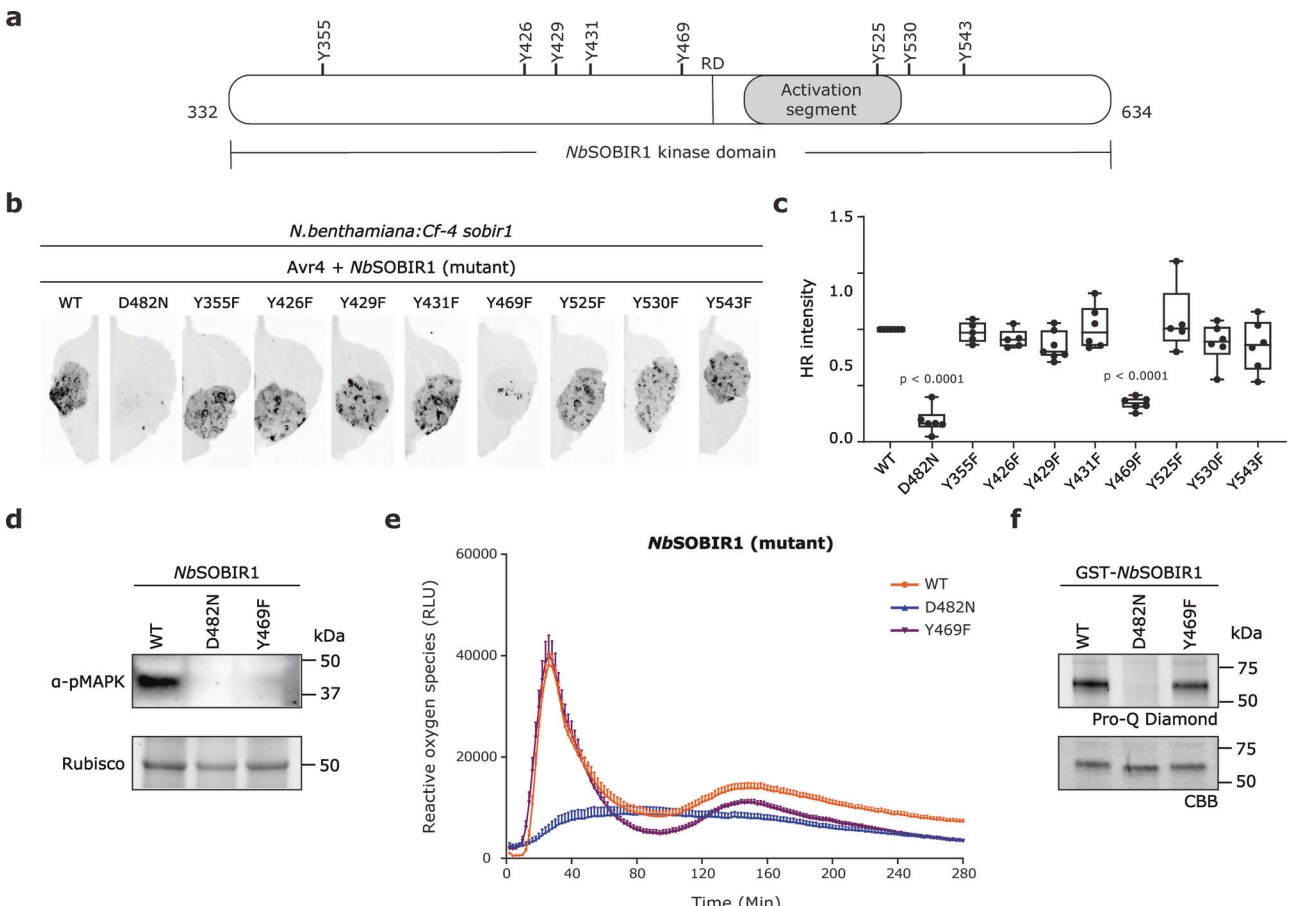

**Fig. 2 | Tyr469 of the kinase domain of SOBIR1 is crucial for the Avr4/Cf-4-triggered HR and MAPK activation, but not for ROS production and intrinsic kinase activity. a** Schematic diagram of the kinase domain of *Nb*SOBIR1, with the location of the activation segment, the RD motif, and all Tyr (Y) residues indicated. **b–d** Complementation with *Nb*SOBIR1 Y469F fails to restore Avr4/Cf-4-triggered HR and MAPK activation in *N. benthamiana:Cf-4 sobir1* knock-out plants. The development of an HR was imaged (**b**) and quantified (**c**) at 5 dpi. Statistical significance was determined by a one-way ANOVA/Dunnett's multiple comparison test, compared with *Nb*SOBIR1 WT. Dots indicate individual values (centre line, median; error bar, minima and maxima; *n* = 6). **e** Transient expression of *Nb*SOBIR1 *Y469F* restores the Avr4/Cf-4-triggered ROS accumulation in *N. benthamiana:Cf-4 sobir1* knock-out plants. Similar to *Nb*SOBIR1 WT, all tested *Nb*SOBIR1 Tyr mutants restored the Avr4/Cf-4-triggered ROS production in this complementation study. Only the results for *Nb*SOBIR1 WT, Y469F and D482N are shown. ROS production is expressed as RLUs, and the data are represented as mean + SEM (*n* = 8). **f** *Nb*SOBIR1 Y469F exhibits intrinsic kinase activity. All experiments were repeated at least three times with similar results, and representative results are shown. Source data are provided as a Source Data file.

mutant was co-expressed with *Avr4* in *N. benthamiana:Cf-4 sobir1* knock-out plants, including *WT* as a positive control and *D482N* as a negative control. Intriguingly, in contrast to the WT, but very similar to the kinase-dead mutant, transient expression of *Nb*SOBIR1 *Y469F* failed to fully complement the Avr4-triggered HR in *N. benthamiana sobir1* knock-out plants (Fig. 2b). Quantification of the HR intensity that was obtained upon complementation by red light imaging[42], showed that the difference in the HR intensity between *Nb*SOBIR1 WT and *Nb*SO-BIR1 Y469F was significant (Fig. 2c). Furthermore, the remaining Tyr mutants showed a similar complementation capacity in *N. benthamiana:Cf-4 sobir1* knock-out plants to the positive control (Fig. 2c).

To further explore the importance of *Nb*SOBIR1 Tyr469 in Avr4/Cf-4-mediated immune signalling, we determined the occurrence of Avr4/Cf-4-induced MAPK activation in *N. benthamiana:Cf-4 sobir1* knock-out plants, after co-expressing *NbSOBIR1 Y469F* with *Avr4*. *Nb*SOBIR1 WT was taken along as a positive control, and the kinase-dead mutant was taken along as a negative control. In contrast to the positive control, but similar to the negative control, co-expression of *NbSOBIR1 Y469F* with *Avr4* failed to restore the Avr4/Cf-4-triggered MAPK activation in *N. benthamiana:Cf-4 sobir1* knock-out plants (Fig. 2d). Again, the lack of MAPK activation was not correlated with its protein accumulation *in planta* (Supplementary Fig. 7).

To investigate whether *Nb*SOBIR1 Tyr469 is also required for regulating the Avr4/Cf-4-triggered ROS burst, we monitored the ROS accumulation in the leaves of *N. benthamiana:Cf-4 sobir1* knock-out plants in which we transiently expressed the individual SOBIR1 mutants, upon adding Avr4 protein. Surprisingly, similar to *Nb*SOBIR1 WT and all other Tyr-to-Phe mutants, transient expression of *Nb*SOBIR1 *Y469F* also partially restored the Avr4-triggered ROS burst in *N. benthamiana:Cf-4 sobir1* knock-out plants (Fig. 2e). These results suggest that *Nb*SOBIR1 Tyr469 is not essential for the Avr4/Cf-4-triggered ROS burst.

Transient expression of *Nb*SOBIR1 *Y469F* at least partially restored the Avr4-triggered ROS production in *N. benthamiana:Cf-4 sobir1* knock-out plants (Fig. 2e), which led us to speculate that this Tyr-to-Phe mutant would still exhibit intrinsic kinase activity. As expected, the recombinant *Nb*SOBIR1 Y469F still showed strong autophosphorylation activity in vitro (Fig. 2f). It is worth noting that we did not observe obvious differences between the staining intensities of the bands upon Pro-Q staining or between the protein mobilities of this mutant and *Nb*SOBIR1 WT (Fig. 2f). Therefore, we conclude that *Nb*SOBIR1 Tyr469 plays no, or only a minor role, in determining the intrinsic kinase activity of *Nb*SOBIR1.

Consistently, the analogous residues of *Nb*SOBIR1 Tyr469 in tomato SOBIR1s, *Sl*SOBIR1 Tyr460 and *Sl*SOBIR1-like Tyr473, gave a

similar phenotype after complementation with their Tyr-to Phe mutants (Supplementary Figs. 6, 7 and 8). Taken together, these observations demonstrate that this particular conserved Tyr residue in the kinase domain of SOBIR1 plays an essential role in mediating specific Avr4/Cf-4-triggered plant immune responses.

**Members of *N. benthamiana* RLCK-VII-6, -7 and -8 are differentially required for the Avr4/Cf-4-triggered biphasic ROS burst**

To search for RLCKs downstream of the Cf-4/SOBIR1/BAK1 complex in Solanaceous plants, we performed a phylogenetic analysis to identify Arabidopsis BIK1 homologues in Arabidopsis, tomato, and *N. benthamiana* (Supplementary Fig. 9a). All the RLCKs were further assigned to six subfamilies, which are referred to as subfamily 4, 5, 6, 7, 8 and 9, and correspond to the RLCK-VII subfamilies in Arabidopsis that were previously reported by Rao and colleagues (Supplementary Fig. 9b)[37]. To identify the RLCKs that play a role in cytoplasmic immune signalling downstream of the activated Cf-4/SOBIR1/BAK1 complex, and to cope with the consequences of functional gene redundancy, we attempted to implement the CRISPR/Cas9 system in *N. benthamiana:Cf-4* to simultaneously target multiple RLCK-encoding genes belonging to the same subfamily (Supplementary Fig. 10 and Supplementary Table 1). *N. benthamiana* RLCK-VII-6 contains 14 members, which might be beyond the limit of the current multiplex CRISPR/Cas9 system (Supplementary Fig. 9b). Hence, the *RLCK* genes for which we had expression data and of which the expression was not down-regulated, were selected to be knocked out. Expression data were obtained from *N. benthamiana:Cf-4* plants also being transgenic for the *Rx* resistance gene against potato virus X (PVX), in which Avr4 and the COAT PROTEIN (CP) of PVX, which matches Rx and also triggers an HR, was transiently expressed. Furthermore, the constitutively active NB-LRR PROTEIN REQUIRED FOR HR-ASSOCIATED CELL DEATH 1 (NRC1) D481V mutant, which triggers an elicitor-independent HR in *N. benthamiana*, was also transiently expressed (Supplementary Fig. 11 and Supplementary Table 1)[47,48].

We performed ROS burst and HR assays on the transgenic plants of the T1 generation to identify the RLCK subfamilies that are required for the Avr4/Cf-4-mediated immune signalling pathway in *N. benthamiana*. As we observed before, the Avr4 protein triggered a biphasic ROS burst in leaf discs obtained from *N. benthamiana:Cf-4* plants[40]. This biphasic ROS burst was generally reduced in all the *rlck-vii-6* and *rlck-vii-7* knock-out lines, and two of the *rlck-vii-8* knock-out lines (Supplementary Fig. 12). In addition, flg22-induced ROS accumulation was dampened to different levels in all the *rlck-vii-6* knock-out lines and was almost eliminated in some of the *rlck-vii-7* and *rlck-vii-8* knock-out lines (Supplementary Fig. 13). This observation indicates that members of RLCK-VII-6, -7, and -8 probably play an important role in regulating the Avr4/Cf-4- and flg22-triggered ROS burst in *N. benthamiana*. Additionally, infiltration of pure Avr4 protein in leaves of *N. benthamiana:Cf-4* induces HR activation, which can be detected and quantified by red light imaging[42]. This HR intensity was significantly attenuated in all the *rlck-vii-7* knock-out lines, but not in the other *rlck-vii* lines (Supplementary Fig. 14), demonstrating that in addition to their role in ROS production, members from RLCK-VII-7 might also be pivotal for the Avr4/Cf-4-triggered HR.

We then selected three independent homozygous *rlck-vii-6* knock-out lines (*rlck-vii-6* #1, *rlck-vii-6* #2, and *rlck-vii-6* #3), two *rlck-vii-7* knock-out lines (*rlck-vii-7* #1 and *rlck-vii-7* #2), and two *rlck-vii-8* knock-out lines (*rlck-vii-8* #1 and *rlck-vii-8* #2) to further verify these results (Supplementary Fig. 15). Of note, these knock-out plants did not display any obvious changes in plant overall morphology, when compared to *N. benthamiana:Cf-4* (Supplementary Fig. 15). Again, we observed that the intensity of the first, early phase of the Avr4/Cf-4-triggered ROS burst was strongly reduced, whereas the second, more

sustained phase of the ROS burst was completely absent in the three independent *N. benthamiana:Cf-4 rlck-vii-6* knock-out lines (Fig. 3a). This observation indicates that members of RLCK-VII-6 play an important role in regulating the Avr4/Cf-4-triggered ROS burst in *N. benthamiana*, especially concerning the second, more sustained phase of the ROS burst. Interestingly, unlike the *rlck-vii-6* knock-out plants, *rlck-vii-7* knock-out plants displayed an overall reduction of the Avr4/Cf-4-triggered ROS burst, and instead of the typical biphasic ROS, only a weak and sustained ROS burst was observed (Fig. 3b). More interestingly, with *rlck-vii-8* knock-out plants, even though the first and second ROS peaks triggered by Avr4/Cf-4 were strongly reduced in size, they were still distinguishable, and additionally, the first ROS peak displayed an obvious delay (Fig. 3c). These results are different from what we observed with the *rlck-vii-6* and *rlck-vii-7* knock-out plants, which demonstrate that there are different downstream ROS regulatory mechanisms and different RLCK-VII subfamilies are employed in different ways to regulate the Avr4/Cf-4-triggered biphasic ROS burst in *N. benthamiana*.

**Members of RLCK-VII-6, -7, and -8 contribute to ROS accumulation in *N. benthamiana* induced by various ExIPs**

Consistent with the aforementioned results, ROS production triggered by the flg22 peptide was strongly compromised in all these knock-out plants, especially in *rlck-vii-7* knock-out plants, when compared to the positive control (Fig. 3d–f). The importance of these RLCK-VII subfamilies in positively regulating ROS accumulation led us to determine their role in regulating ROS burst induced by other ExIPs, such as chitin. Strikingly, all the independent knock-out lines also exhibited an obvious reduction in chitin-triggered ROS production, especially *rlck-vii-8* knock-out lines, when compared to the positive control (Fig. 3g–i). The Arabidopsis LRR-RLPs RLP23 and RLP42 perceive a conserved 20-amino-acid peptide from necrosis and ethylene-inducing peptide 1 (NEP1)-like proteins (nlp20) and *Botrytis cinerea* endo-polygalacturonases (PGs and their derived peptide, pg13), respectively[20,49,50]. Similar to Cf-4, both RLP23 and RLP42 require SOBIR1 and BAK1 to initiate immune signalling upon recognition of their matching elicitor[14,20,49,50]. Interestingly, *N. benthamiana* plants that overexpress either *RLP23* or *RLP42* show sensitivity to their corresponding ExIPs. To decipher whether members of RLCK-VII-6, -7, and -8 also play a role downstream of RLP23 and RLP42, we transiently overexpressed them in the different knock-out lines, followed by adding the corresponding ligands and monitoring ROS production. Interestingly, similar to the situation with Avr4/Cf-4, both the nlp20/RLP23- and pg13/RLP42-mediated ROS burst in *N. benthamiana* plants was biphasic, of which the first burst was rapid with a relatively high amplitude, whereas the second burst was sustained, with relatively low amplitude (Supplementary Fig. 16). Both the nlp20/RLP23- and the pg13/RLP42-triggered ROS production was significantly impaired in all knock-out plants, when compared to the positive control (Supplementary Fig. 16). This was especially the case for the second burst in all *rlck-vii-6* knock-out plants, which was completely abolished (Supplementary Fig. 16a, b). Collectively, these results indicate that the members of RLCK-VII-6, -7, and -8 are required for regulating the ROS production triggered by a broad spectrum of ExIPs in *N. benthamiana*.

**Members of RLCK-VII-7 play an important role in the Avr4/Cf-4-induced HR and in resistance of *N. benthamiana* to *Phytophthora palmivora***

In addition to ROS production, the activation of a downstream MAPK cascade is another key signalling event of ExTI[43,51]. We observed that infiltration of the Avr4 protein in leaves of *N. benthamiana:Cf-4* induces the activation of a MAPK cascade within five minutes (Supplementary Fig. 17). To explore whether members of RLCK-VII-6, -7, and -8 are also involved in Avr4/Cf-4-triggered MAPK activation, we infiltrated pure Avr4 protein in leaves of the different knock-out lines as well as in

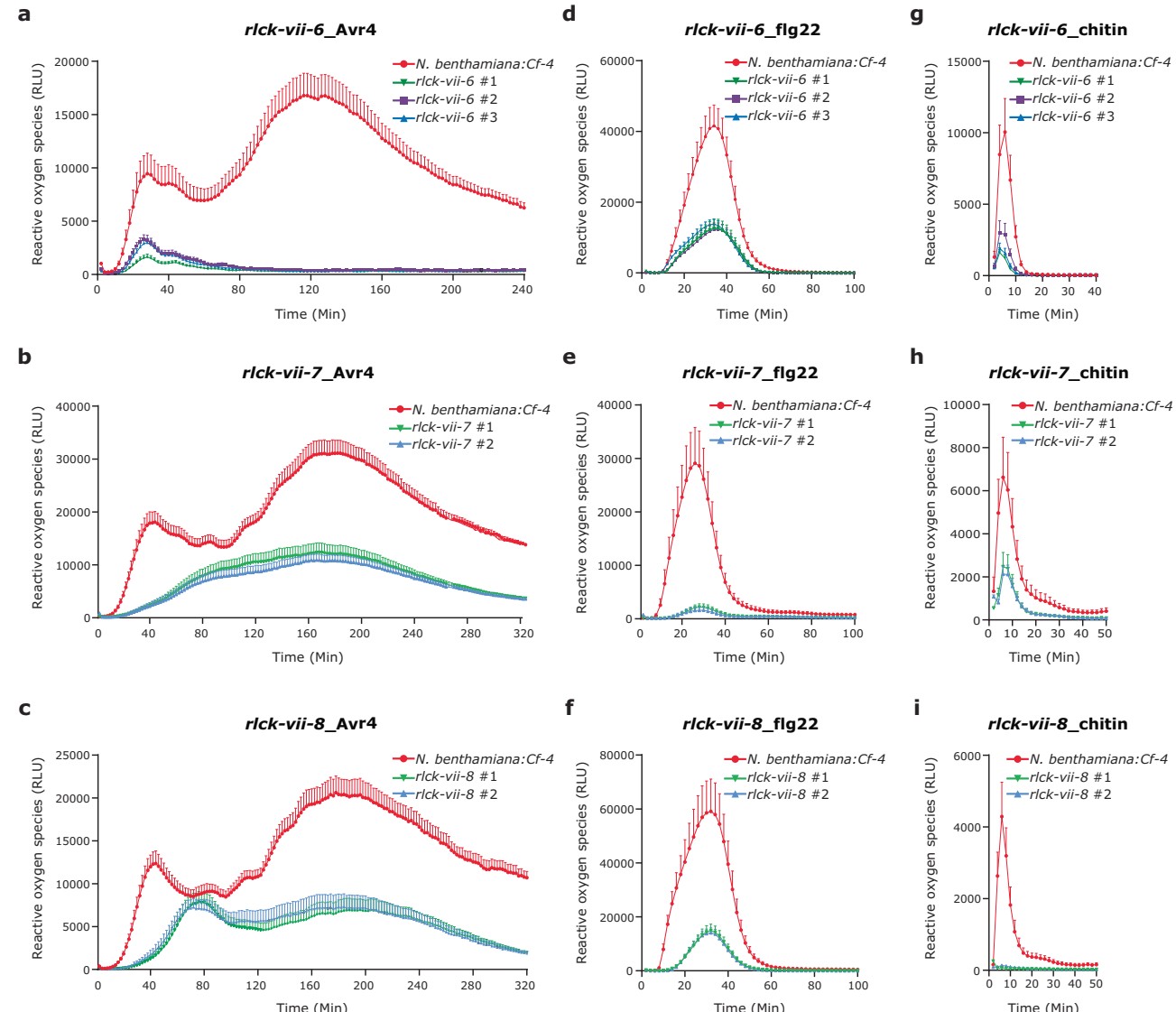

**Fig. 3 | Members of RLCK-VII-6, -7, and -8 differentially contribute to ROS accumulation in *N. benthamiana* induced by various ExIPs.** *N. benthamiana:Cf-4 rlck-vii-6, rlck-vii-7,* and *rlck-vii-8* homozygous knock-out lines show a reduced ROS production when compared to *N. benthamiana:Cf-4* (the positive control), after treatment with 0.1 μM Avr4 protein (**a**–**c**), 0.1 μM flg22 (**d**–**f**), and 10 μM chitin (**g**–**i**). ROS production is expressed as RLUs, and the data are represented as mean + SEM (*n* = 12 for Avr4 treatment, *n* = 8 for flg22 and chitin treatments). All experiments were repeated at least three times with similar results, and representative results are shown. Source data are provided as a Source Data file.

leaves of *N. benthamiana:Cf-4* plants. The leaf samples were harvested at 15 min after Avr4 infiltration, followed by performing a western blot assay revealing MAPK activation. Intriguingly, no obvious changes in MAPK activation in all the knock-out lines were observed at the tested time point when compared to *N. benthamiana:Cf-4*, suggesting that these RLCK members are dispensable for the Avr4/Cf-4-triggered MAPK activation (Fig. 4a–c).

Ethylene is an important signalling molecule that orchestrates plant defence against microbial pathogens. We then examined whether this ethylene production is affected by knocking out the different RLCK-VII subfamilies. For this, independent homozygous *N. benthamiana:Cf-4 rlck-vii-6, rlck-vii-7,* and *rlck-vii-8* knock-out lines, as well as *N. benthamiana:Cf-4*, were treated with 2.5 μM Avr4 protein, followed by measuring ethylene accumulation. We observed that Avr4 triggers a strong ethylene production in *N. benthamiana:Cf-4*. The different knock-out lines displayed no significant alterations in ethylene production (Fig. 4d–f), indicating that these RLCKs play no, or only a minor role, in regulating the Avr4/Cf-4-induced ethylene production. It is worth

noting that the *rlck-vii-8* knock-out lines showed a slightly increased ethylene production, implying that members of RLCK-VII-8 might be inhibitors of ethylene signalling, triggered by Avr4/Cf-4 (Fig. 4f).

To verify that members of RLCK-VII-7 are required for the Cf-4/ Avr4-triggered HR in *N. benthamiana*, we infiltrated pure Avr4 protein in leaves of these independent homozygous knock-out lines and of *N. benthamiana:Cf-4* as a control. Consistent with what we observed before, Avr4 protein infiltration revealed significant changes in the capacity to mount an Avr4/Cf-4-triggered HR of the *rlck-vii-7*, but not the *rlck-vii-6* and *rlck-vii-8* knock-out plants, when compared to *N. benthamiana:Cf-4* (Fig. 4g–i). These results suggest that members of RLCK-VII-7 play an essential role in regulating the Avr4/Cf-4-triggered HR.

To investigate the actual role of these RLCKs in host defence against microbial pathogens, we inoculated leaves from all the independent *rlck* knock-out lines with the oomycete pathogen *Phytophthora palmivora*. We used *N. benthamiana:Cf-4 sobir1* as a positive control, as SOBIR1 was already shown earlier to be required for resistance of tomato and *N. benthamiana* to several *Phytophthora*

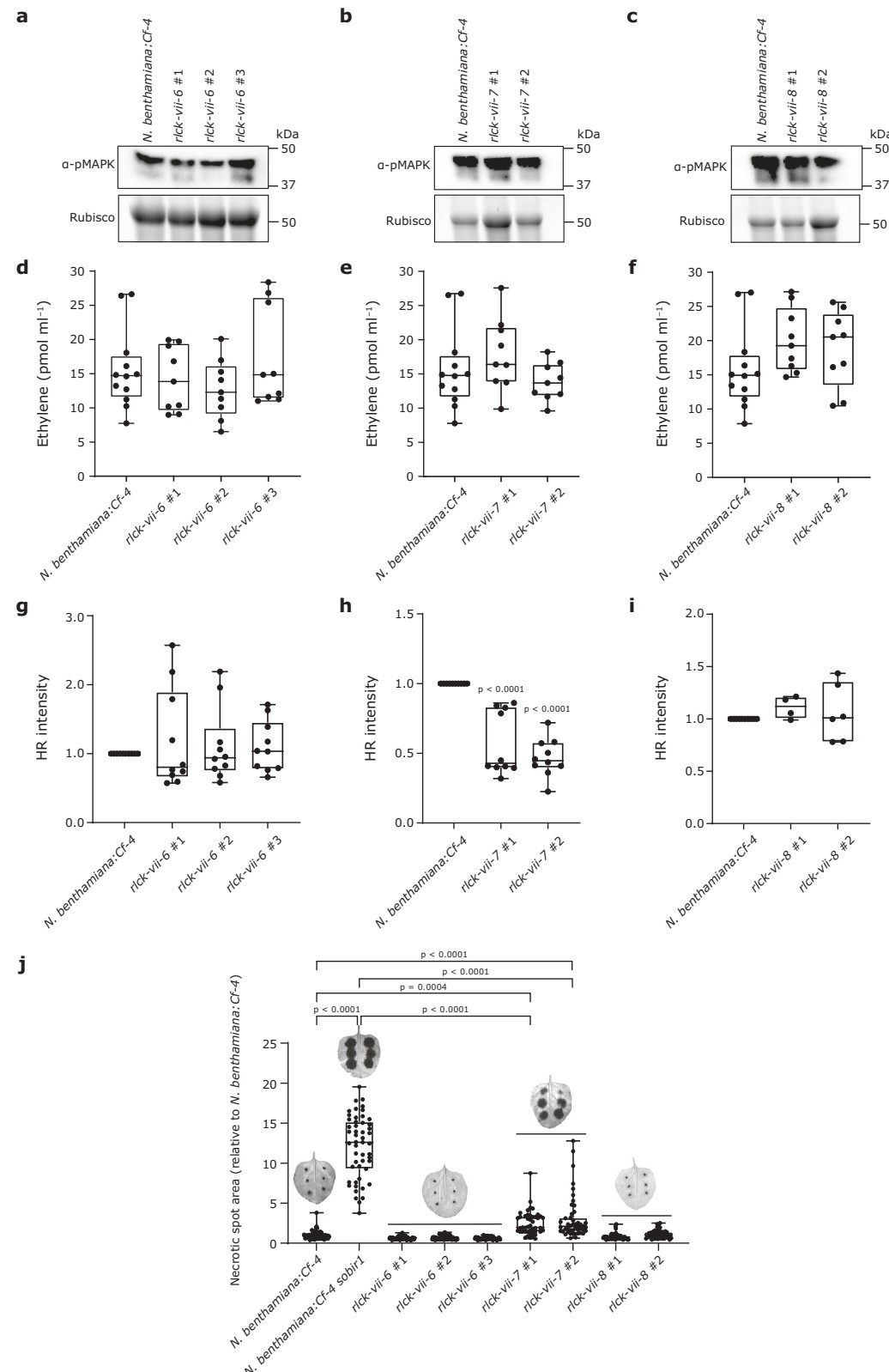

species[52,53]. In agreement with these earlier observations, *N. benthamiana:Cf-4 sobir1* plants were indeed more susceptible to *P. palmivora* (Fig. 4j). Interestingly, we found that *N. benthamiana:Cf-4 rlck-vii-7* knock-out plants were also more susceptible to *P. palmivora* infection than *N. benthamiana:Cf-4* and the other *rlck-vii* knock-out plants, but to a lower extent than the *N. benthamiana:Cf-4 sobir1* plants (Fig. 4j). These observations demonstrate that SOBIR1 and members of RLCK-

VII-7 contribute to the immune response of *N. benthamiana* to *P. palmivora*.

## SOBIR1 and BAK1 directly phosphorylate members of RLCK-VII-6, -7 and -8 in vitro

We next asked whether these important RLCK members might be trans-phosphorylated by the activated Cf-4/SOBIR1/BAK1 complex.

**Fig. 4 | Members of RLCK-VII-7 play an important role in the Avr4/Cf-4-induced HR and host resistance against *Phytophthora palmivora*.** *N. benthamiana:Cf-4 rlck-vii-6, rlck-vii-7*, and *rlck-vii-8* knock-out plants do not exhibit an altered Avr4/Cf-4-triggered MAPK activation (**a**–**c**) and ethylene production (**d**–**f**), when compared to *N. benthamiana:Cf-4* plants. **a**–**c**, Leaf samples were harvested at 15 min after infiltration with 5 μM Avr4 protein. Total protein extracts were run on SDS gel and subjected to immunoblotting employing a p42/p44-erk antibody specifically detecting phosphorylated, and thereby activated, MAPKs (α-pMAPK). **d**–**f** Ethylene production by the various *rlck* knock-out plants was measured at 4 h after treatment with 2.5 μM Avr4 protein. Data points are indicated as dots from three independent experiments and plotted as box plots (centre line, median; error bar, minima and maxima; one-way ANOVA/Dunnett's multiple comparison test; $n = 9$). **g**–**i** The HR intensity triggered by Avr4 (5 μM) in leaves of *rlck-vii-7* knock-out plants

is significantly reduced when compared to the intensity of the HR in leaves of *N. benthamiana:Cf-4* plants. The HR was imaged by the Chemidoc and the HR intensity was quantified by Image Lab, at 2 dpi. Statistical significance was determined by a one-way ANOVA/Dunnett's multiple comparison test, compared with *N. benthamiana:Cf-4*. Dots indicate individual values (centre line, median; error bar, minima and maxima; $n = 6$). **j** Knocking out *SOBIR1* or members of RLCK-VII-7 enhances *N. benthamiana* susceptibility to *P. palmivora*. Lesion areas on leaves of 5-week-old *N. benthamiana* plants were quantified at 2 days post inoculation. Values are given relative to *N. benthamiana:Cf-4*. Statistical significance was assessed using one-way ANOVA with Tukey's HSD (Honestly Significant Difference) post-hoc test. Dots indicate individual values ($N = 45$). Letters indicate statistical groupings. Experiments were repeated three times with similar results. Source data are provided as a Source Data file.

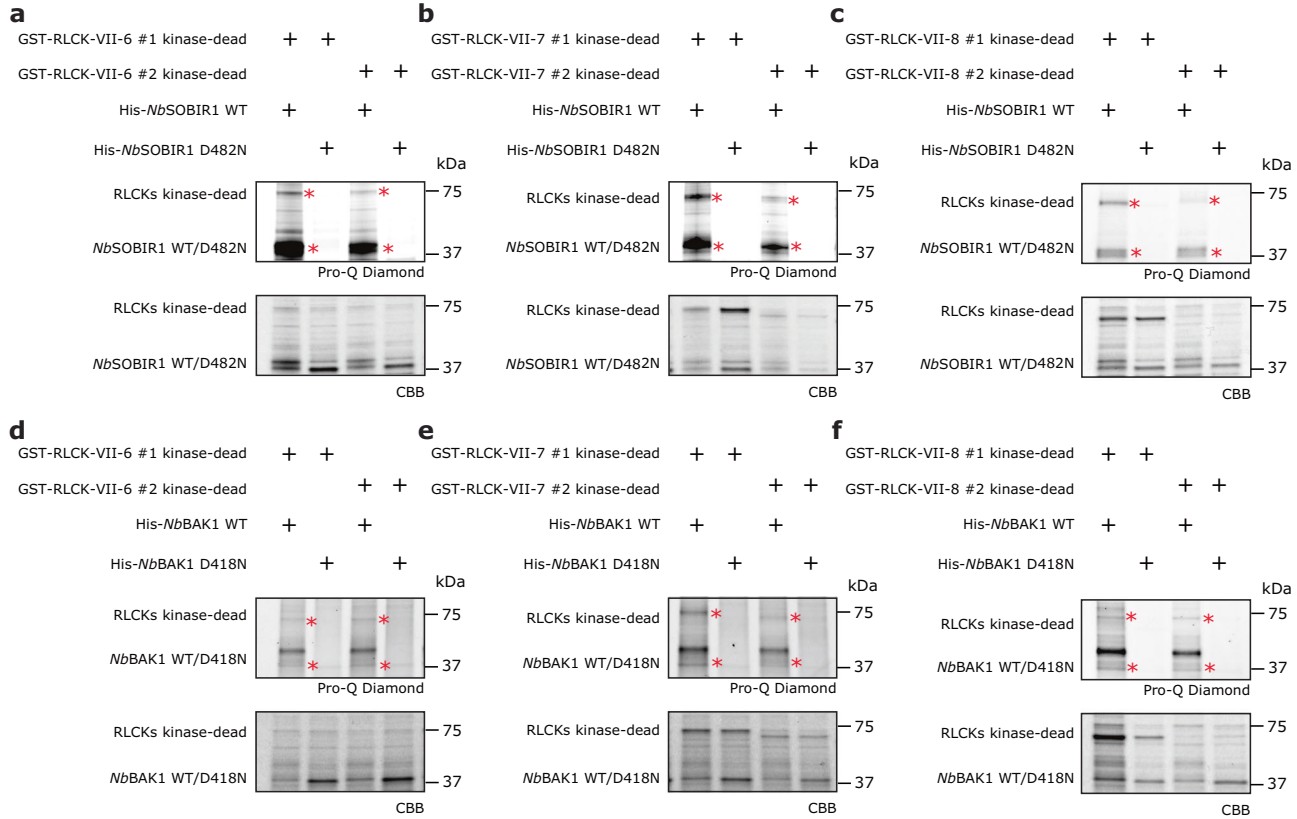

**Fig. 5 | Members of *N. benthamiana* RLCK-VII-6, −7, and −8 are directly trans-phosphorylated by both SOBIR1 and BAK1 in vitro.** Two members were randomly selected from RLCK-VII-6, -7, and -8 and their kinase-dead mutants were co-expressed with either the cytoplasmic kinase domain from *Nb*SOBIR1 WT or its D482N kinase-dead mutant (**a**–**c**), or with either the cytoplasmic kinase domain from *Nb*BAK1 WT or its D418N kinase-dead mutant (**d**–**f**), in *E. coli*. After SDS-PAGE of the boiled cell lysate, the phosphorylation status of the recombinant proteins was determined by performing a Pro-Q Diamond stain (top panels), while the total proteins were stained with CBB (bottom panels). Bands with the expected sizes are

indicated with a red asterisk. Experiments were repeated two times with similar results, and representative results are shown. RLCK-VII-6 #1 kinase-dead, Niben101Scf02460g01004 K110A; RLCK-VII-6 #2 kinase-dead, Niben101Scf06739g05004 K110A; RLCK-VII-7 #1 kinase-dead, Niben101Scf00712g13012 K127A; RLCK-VII-7 #2 kinase-dead, Niben101Scf01176g01025 K119A; RLCK-VII-8 #1 kinase-dead, Niben101Scf00012g00012 K109A; RLCK-VII-8 #2 kinase-dead, Niben101Scf06482g03003 K114A. Source data are provided as a Source Data file.

Both SOBIR1 and BAK1 exhibit strong trans-phosphorylation activity (Fig. 1 and Supplementary Fig. 4), which provides the possibility that their kinase domains might be able to directly trans-phosphorylate these RLCKs. To verify this hypothesis, an in vitro phosphorylation assay was conducted.

Two members were randomly selected from each of the RLCK-VII-6, -7, or -8 subfamilies, and their kinase-dead mutants were co-expressed with either the cytoplasmic kinase domain of *Nb*SOBIR1 WT or *Nb*BAK1 WT in *E. coli*, with *Nb*SOBIR1 D482N (a kinase-dead mutant of *Nb*SOBIR1) or *Nb*BAK1 D418N (a kinase-dead mutant of *Nb*BAK1), as a negative control. The proteins were run on SDS-PAGE gels and a CBB

stain showed that all proteins accumulated properly in vitro (Fig. 5a–f). In line with the aforementioned observations, both *Nb*SOBIR1 WT and *Nb*BAK1 WT showed strong auto-phosphorylation, while their kinase-dead mutants did not. Importantly, no phosphorylated proteins were observed when the kinase-dead mutants of the selected RLCK members were co-expressed with either *Nb*SOBIR1 D482N or *Nb*BAK1 D418N, while these RLCKs did get phosphorylated when being co-expressed with *Nb*SOBIR1 WT and *Nb*BAK1 WT (Fig. 5a–f). These results demonstrate that both SOBIR1 and BAK1 indeed directly trans-phosphorylate members from the RLCK-VII subfamilies -6, -7 and -8, and that these trans-phosphorylation events rely on the kinase activity

of SOBIR1 and BAK1. Likely, the trans-phosphorylation of these RLCKs by SOBIR1 and BAK1 is required for the activation of these RLCKs and subsequent initiation of the downstream signalling pathway.

## SOBIR1 interacts with various RLCKs *in planta*

To determine whether the tomato Cf-4/SOBIR1 complex actually interacts with RLCKs *in planta*, we tested the ability of several RLCKs from tomato to interact with *Sl*SOBIR1 using a split-luciferase assay[54]. We focussed on 10 tomato *RLCKs*, of which the expression is at least three times upregulated at three days after inoculation of leaflets of tomato cv. Moneymaker with the necrotrophic fungal pathogen *B. cinerea*, as compared to mock-inoculated leaflets[55]. Cluc-tagged RLCKs, as well as *At*BIK1-Cluc, were analysed for their ability to interact with Nluc-tagged *Sl*SOBIR1 *in planta* (Supplementary Fig. 18a–l). GUS-Nluc was included as a negative control. To infer interaction, we focused on the presence of increased luminescence signals when the RLCKs were co-expressed with *Sl*SOBIR1, as compared to GUS, in *N. benthamiana*. All 10 Cluc-tagged RLCKs were found to properly accumulate in leaves of *N. benthamiana* upon their transient expression, albeit at different levels (Supplementary Fig. 18m).

The results of the split-luciferase assay (Supplementary Fig. 18a–l) reveal that Solyc07g041940 (*Sl*41940) (RLCK-VII-6), Solyc11g062400 (*Sl*62400) (RLCK-VII-9), Solyc01g088690 (*Sl*88690) (RLCK-VII-4), Solyc05g007050 (*Sl*7050) (RLCK-VII-5), and Solyc08g077560 (*Sl*77560) (RLCK-VII-4) do not specifically interact with *Sl*SOBIR1, as the luminescence signals did not really differ from the signals obtained upon their co-expression with the GUS control (Supplementary Fig. 18a, c, e, h–j). On the contrary, Solyc04g082500 (*Sl*82500) (RLCK-VII-7), Solyc06g062920 (*Sl*62920) (RLCK-VII-6), Solyc05g025820 (*Sl*25820) (RLCK-VII-6), Solyc01g112220 (*Sl*112220) (RLCK-VII-7), and Solyc06G005500 (*Sl*05500) (RLCK-VII-8) likely do specifically interact with *Sl*SOBIR1, similar to *At*BIK1 (Supplementary Fig. 18a, d, f, g, k, l). *Sl*25820 clusters with *At*PBL13 and *At*RIPK (Supplementary Fig. 9), which are RLCKs known to be involved in defence[34,56]. *Sl*62920 is a homologue of *Sl*ACIK1 (Supplementary Fig. 9), which has been shown before to play a role in Cf-mediated immunity to *F. fulva*[57]. *Sl*82500 and Sl112220 in their turn are closely related to Arabidopsis PBL30, PBL31 and PBL32, of which PBL30 (also known as CAST AWAY, CST) plays an important role in RLP1/SOBIR1-, RLP23/SOBIR1-, and RLP42/SOBIR1-mediated immunity (Supplementary Fig. 9)[36]. Both PBL30 and PBL31 have indeed been shown to interact with SOBIR1/EVR[36,58].

Having shown, by a split-luciferase assay in *N. benthamiana*, that several tomato RLCKs interact *in planta* with *Sl*SOBIR1, we decided to use an alternative approach to test for interaction between *Nb*SOBIR1 and endogenous RLCKs from *N. benthamiana*. For this, we performed a series of TurboID (TbID)-based proximity-dependent labelling (PL)-mass spectrometry (MS) experiments[59], for which we transiently expressed the fusion proteins *Nb*SOBIR1-YFP-TbID and the control GUS-YFP-TbID in *N. benthamiana*:Cf-4 sobir1 knock-out plants[40].

Interestingly, we identified several peptides of biotinylated RLCKs, meaning that these proteins are in close proximity to the kinase domain of *Nb*SOBIR1 (Supplementary Fig. 19). Strikingly, the RLCK-VII-8 member Niben101Scf00012g00012, being the homologue of tomato Solyc06g005500 (*Sl*05500) that seems to specifically interact only with *Sl*SOBIR1 in the split-luciferase assay (Supplementary Fig. 18a, l), was also identified. Eventually, we identified four distinct members of RLCK-VII-8, one member of RLCK-VII-6 and peptides matching RLCKs of classes IV and VIII for which we were not able to assign them to distinct subfamily members (Supplementary Fig. 19).

## A complementation assay shows that transient expression of RLCK *Niben101Scf00012g00012* partially restores the ROS burst

Having observed that several RLCKs interact with *Nb*SOBIR1 in *N. benthamiana*, we asked whether RLCKs belonging to one particular subfamily are redundant, allowing the transient expression of one of them to restore the biphasic ROS burst, or whether each of them has a non-redundant function, requiring the expression of all subfamily members at the same time to restore this biphasic ROS burst. We chose to perform a complementation assay with the RLCK-VII-8 member Niben101Scf00012g00012 in the *rlck-vii-8* #1 knock-out line and observed that transient expression of this RLCK resulted only in a partial restoration of the ROS burst, as a proper biphasic ROS was not observed (Supplementary Fig. 20). This result suggests that all members of a particular RLCK subfamily might have overlapping redundant functions in triggering ROS production, but each of them also has a specific role in mounting this part of the defence response of the plant.

## Discussion

SOBIR1 and BAK1 are well-known co-receptors for LRR-RLPs, such as Cf-4, RLP23, and RLP42[14,15,18,20,50]. It has been reported that an RLP/SOBIR1 complex forms heterodimers with BAK1 upon its elicitation and that subsequent trans-phosphorylation events between the kinase domains of SOBIR1 and BAK1 are required for initiating downstream immune signalling[41]. Both SOBIR1 and BAK1 are RD kinases, suggesting the possibility of auto-phosphorylation in their activation segment. Here, our in vitro studies demonstrate that, similar to their Arabidopsis homologues, *Nb*SOBIR1, *Sl*SOBIR1, *Sl*SOBIR1-like, *Nb*BAK1 and *Sl*BAK1 exhibit auto-phosphorylation activity (Fig. 1 and Supplementary Fig. 4)[38,60,61]. Furthermore, SOBIR1 indeed can directly phosphorylate BAK1, and the intrinsic kinase activity of SOBIR1 is required for this trans-phosphorylation event (Figs. 1g and 6, Supplementary Fig. 4c, d). Accordingly, BAK1 is also able to directly phosphorylate SOBIR1, which again depends on the kinase activity of BAK1 (Figs. 1h and 6, and Supplementary Fig. 4e, f). Interestingly, our observations are supported by a recent study, which shows the auto-phosphorylation of *At*SOBIR1 via an intermolecular mechanism, and the trans-phosphorylation between *At*SOBIR1 and *At*BAK1 in vitro[62]. Previously, we have reported that *At*SOBIR1, which constitutively activates immune responses when overexpressed in *N. benthamiana*, is highly phosphorylated. Kinase activity of both *At*SOBIR1 and *At*BAK1 is required for both the *At*SOBIR1-induced constitutive cell death response and for the phosphorylation of the kinase domain of *At*SOBIR1[41]. In addition, transient co-expression of *Avr4* with the dominant negative BAK1 mutants *AtBAK1C408Y* or *AtBAK1D416N* in *N. benthamiana:Cf-4* results in a reduced Avr4/Cf-4-triggered HR[41]. Therefore, we propose that trans-phosphorylation events between the cytoplasmic kinase domains of SOBIR1 and BAK1 eventually result in the full activation of SOBIR1/BAK1-containing immune complexes, which is essential for LRR-RLP-mediated immunity. As many LRR-RLPs that are involved in plant immunity require SOBIR1 and BAK1 for their function, the model that we proposed earlier[41], and is now further supported by this study and by the work of Wei and colleagues[62], probably also applies to immune signalling triggered by additional RLP/SOBIR1 complexes. Moreover, increasing evidence has indicated that BAK1 promotes the activation of the receptor complex upon the perception of an ExIP by the LRRs of the matching primary receptor. However, the signalling specificity is determined by the kinase domain of the primary receptor, which is either an RLK or the constitutive RLP/SOBIR1 complex, whereas BAK1 merely acts as a general complex activator[63]. Therefore, after being trans-phosphorylated by BAK1, SOBIR1 is proposed to initiate the trans-phosphorylation events with downstream cytoplasmic signalling components. Such components are for example particular RLCKs, thereby triggering a specific type of immune signalling, irrespective of the RLP that is involved in the RLP/SOBIR1 complex.

Furthermore, in this study, we show that *Nb*SOBIR1 Thr522 and its analogous residues in tomato SOBIR1s (*Sl*SOBIR1 Thr513 and *Sl*SOBIR1-like Thr526), present in the activation segments of SOBIR1, are essential for their intrinsic kinase activity and thereby for the initiation of the

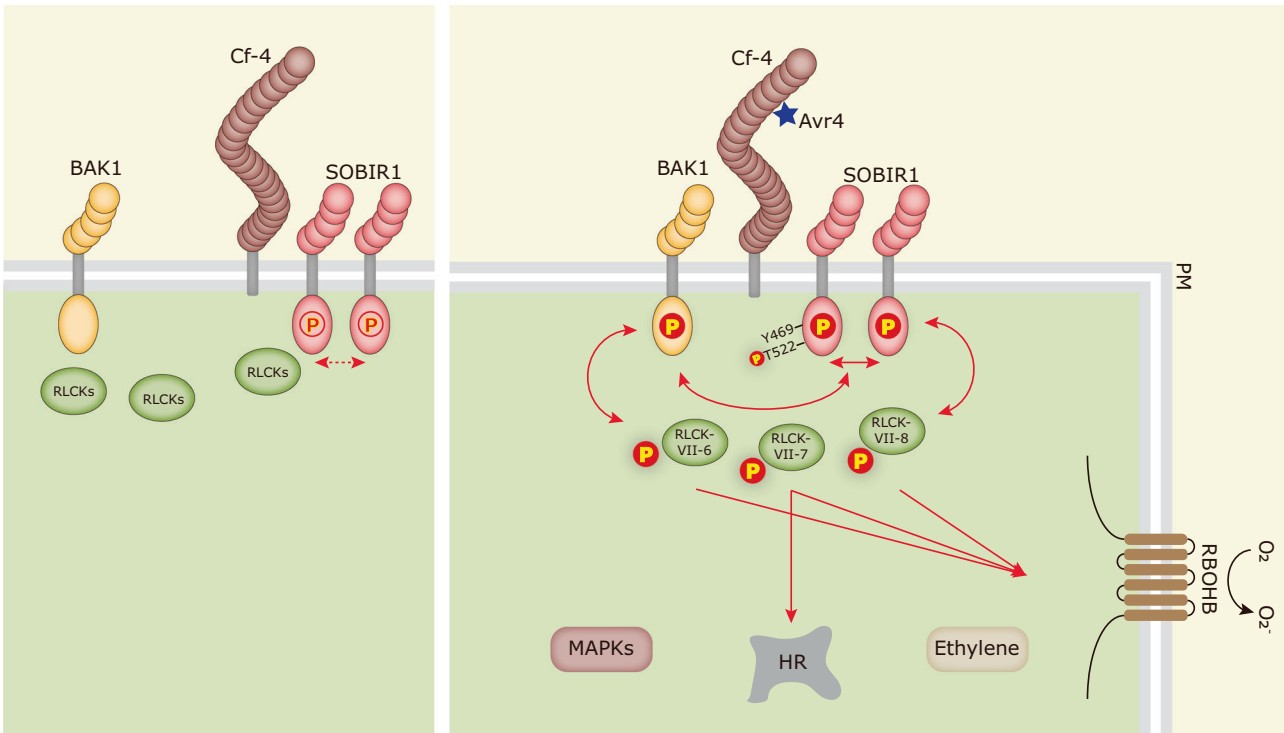

**Fig. 6 | Working model for the activation of the Cf-4/SOBIR1/BAK1 immune complex, and the initiation of downstream immune signalling by the activated complex.** In the absence of the Avr4 effector (left panel), the resistance protein Cf-4 constitutively interacts with SOBIR1. On the other hand, SOBIR1 constitutively forms homodimers that allow basal activation of the SOBIR1 kinase domain through cross-phosphorylation. In response to the perception of Avr4 by Cf-4 (right panel), SOBIR1 initiates strong auto-phosphorylation of its kinase domain, and the amino acid residue Thr522 is required for its intrinsic kinase activity. Meanwhile, the Cf-4/SOBIR1 complex recruits BAK1, after which trans-phosphorylation events between the cytoplasmic kinase domains of SOBIR1 and BAK1 take place. Members of the *N. benthamiana* RLCK-VII-6, 7, and 8 are required for the Avr4/Cf-4-triggered ROS burst, whereas members of RLCK-VII-7 are also required for the Avr4/Cf-4-induced HR. SOBIR1 Tyr469 plays an essential role in the Avr4/Cf-4-triggered HR and MAPK activation, but not in ROS accumulation and SOBIR1 intrinsic kinase activity. As this particular residue is predicted to be solvent-exposed, it is proposed that this residue regulates plant immune responses by interacting with specific downstream signalling partners. All selected RLCK-VII members can be directly trans-phosphorylated by both SOBIR1 and BAK1. Likely, these trans-phosphorylation events lead to the activation of these RLCKs, after which they can phosphorylate the RBOHB oxidase, leading to the accumulation of apoplastic ROS. Members of RLCK-VII-7 possibly phosphorylate various transcription factors or other downstream signalling components to trigger the activation of the HR. Solid arrows indicate signalling events with supporting data presented in this study, whereas dashed arrows indicate proposed events. The red open (left panel) and filled (right panel) circles with a 'P' inside represent low levels and high levels of phosphorylation, respectively. PM plasma membrane.

Avr4/Cf-4-mediated immune responses (Fig. 1, and Supplementary Figs. 2 and 4). These results are further supported by a recent study, which shows that *At*SOBIR1 T529A does not exhibit intrinsic kinase activity, thereby resulting in a complete loss of cell death in *N. benthamiana* upon its overexpression[62]. Interestingly, the equivalent Thr residue has been proven essential for many other RD kinases. A good example is Arabidopsis RLK CHITIN ELICITOR RECEPTOR KINASE1 (CERK1) Thr479 (Supplementary Fig. 1), which is indispensable for the activation of the *At*CERK1 kinase[64,65]. Furthermore, the Arabidopsis LRR-RLK NUCLEAR SHUTTLE PROTEIN (NSP)-INTERACTING KINASE1 (NIK1) is a virulence target of the begomovirus NSP and is involved in plant antiviral immunity[66,67]. A mutation at Thr474, which is located at the activation segment of the kinase domain of NIK1 (Supplementary Fig. 1), attenuates the auto-phosphorylation of NIK1 and enhances susceptibility to the Cabbage leaf curl virus[68]. Therefore, this particular Thr residue might play a general role in regulating the intrinsic kinase activity of RD kinases.

In addition to Thr522, Tyr469 (as well as its analogous residues in *Sl*SOBIR1 and *Sl*SOBIR1-like) in the kinase domain of *Nb*SOBIR1 has also been identified to be crucial for the Avr4/Cf-4-induced HR and MAPK activation, but not for its intrinsic kinase activity and Avr4/Cf-4-triggered ROS accumulation (Fig. 2, Supplementary Figs. 6 and 8). Strikingly, a vital role in regulating plant immunity has recently been assigned to this particular Tyr residue present in the kinase domain of

several well-known RLKs[69,70–73]. For instance, upon the perception of elf18, the Arabidopsis LRR-RLK EFR phosphorylates at Tyr836, which is equivalent to *Nb*SOBIR1 Tyr469, and this phosphorylation is required for the activation of EFR itself and the initiation of sequential downstream immune responses[70]. Arabidopsis CERK1 is the co-receptor of the fungal cell wall component chitin[74,75]. Phosphorylation of CERK1 Tyr428, which is also analogous to *Nb*SOBIR1 Tyr469, is essential for chitin-triggered CERK1 activation, ROS production, MAPK activation, downstream RLCK phosphorylation, and resistance to the fungal pathogen *B. cinerea*[69]. These results collectively demonstrate the importance of this particular Tyr residue in the kinase domain of cell-surface RLKs that are involved in plant immunity. Although *At*SOBIR1 has been reported to auto-phosphorylate at Tyr residues[38], no phosphorylated Tyr residues have been detected by mass spectrometry (MS) in the kinase domain of *At*SOBIR1, when auto-phosphorylated, or after being trans-phosphorylated by BAK1 in vitro[62], or when produced *in planta*[76]. This is in contrast to the aforementioned RLKs, which can phosphorylate at this Tyr residue. Nonetheless, based on the structure of the *At*SOBIR1 kinase domain, which has been determined recently[62], *At*SOBIR1 Tyr476 (equivalent Tyr residue of *Nb*SOBIR1 Tyr469) is solvent-exposed, therefore, it might be easier to access by downstream components or other regulatory proteins (Supplementary Fig. 21). Therefore, this important Tyr residue in the kinase domain of SOBIR1 might not regulate plant immune responses by phosphorylation, as the

Tyr469Phe mutant of SOBIR1 is still kinase-active, but by interacting with specific downstream signalling partners, such as members of RLCK-VII-7 that are required for Avr4/Cf-4-induced HR (Fig. 4h).

We observed that the kinetics of the Avr4-induced biphasic ROS burst displayed differently in *N. benthamiana:Cf-4 rlck-vii-6, rlck-vii-7* and *rlck-vii-8* (Fig. 3a–c). For the *rlck-vii-6* knock-out lines, the second burst is specifically and completely inhibited (Fig. 3a); for the *rlck-vii-7* knock-out lines, the overall ROS production is strongly attenuated, and only a weak and sustained ROS burst is triggered (Fig. 3b); whereas, for the *rlck-vii-8* knock-out lines, both ROS bursts are strongly compromised, with the first burst being delayed (Fig. 3c). This raises the possibility that there are different downstream ROS regulatory mechanisms in *N. benthamiana*, which together determine the ROS profile. This option is supported by our observation that various RLCKs belonging to different subfamilies interact with the kinase domain of SOBIR1 *in planta* (Supplementary Figs. 18 and 19) and that these different RLCKs have non-redundant functions (Supplementary Fig. 20). In Arabidopsis, RESPIRATORY BURST OXIDASE HOMOLOG D (RBOHD) is engaged in extracellular ROS production, and growing evidence has suggested that RLCKs differentially regulate RBOHD activation through the differential phosphorylation of various sites in the RBOHD enzyme[77]. For instance, BIK1 directly phosphorylates the N-terminus of RBOHD to positively regulate ROS production in Arabidopsis, whereas PBL13, which negatively regulates RBOHD activation, directly phosphorylates the C-terminus of RBOHD. In *N. benthamiana*, the RBOHB homologue is responsible for the fast apoplastic ROS production during the establishment of immunity, and therefore we speculate that members from RLCK-VII-6, RLCK-VII-7 and RLCK-VII-8 phosphorylate RBOHB at different sites, thereby causing different ROS kinetics (Fig. 6)[78]. Furthermore, it has been reported that in *N. benthamiana* the first burst of the biphasic apoplastic ROS burst is mediated by swift RBOHB phosphorylation, whereas the second burst is the result of transcriptional upregulation of the *RBOHB* gene, a process that is mediated by activated WRKY transcription factors[79]. Hence, we hypothesize that members from RLCK-VII-6, upon their activation by the upstream cell-surface complex, phosphorylate RBOHB at specific sites for the swift ROS burst, and meanwhile directly or indirectly phosphorylating certain transcription factors to regulate the later phase of the ROS burst.

To determine the role of studied RLCKs in the resistance response of *N. benthamiana* to microbial pathogens, we challenged *sobir1* knock-out plants and all the *rlck* knock-out plants with the oomycete pathogen *P. palmivora*. Our results demonstrate that SOBIR1 and members of RLCK-VII-7, the latter playing an essential role in positively regulating both the Avr4/Cf-4-triggered ROS burst and the HR, are all involved in immunity to *P. palmivora*, with SOBIR1 playing a more important role than all members of RLCK-VII-7 together (Fig. 4j). Interestingly, Liang and co-workers[80] performed a similar experiment in Arabidopsis, inoculating higher-order *rlck-vii* mutants with *P. capsici*. They found that RLCK-VII-6 and RLCK-VII-8 members are required for resistance to this oomycete pathogen. This observation suggests that in different plant species the various RLCK-VII subfamilies play different roles in immunity. Previous studies have shown that some ExIPs (for example INF1 from *P. infestans*, XEG1 from *P. sojae*, and ParA1 from *P. parasitica*), are perceived by RLPs of *N. benthamiana* and that the HR that is triggered requires both SOBIR1 and BAK1[52,53,81,82]. Therefore, it is very likely that also here an RLP/SOBIR1 complex, which forms the frontline of plant innate immunity, plays a prominent role in warding off invasion by *P. palmivora*. Indeed, it has been shown that this oomycete pathogen produces conserved elicitins that are highly homologous to for example INF1[83] and can potentially be recognised by the endogenous *N. benthamiana* LRR-RLP that is referred to as RESPONSIVE TO ELICITINS (REL)[82]. The various members of RLCK-VII-7 are probably, partially redundant with RLCK-VII-6 and -8 members, involved in transducing immune signalling from the activated RLP/SOBIR1/BAK1 complex to downstream signalling partners. Understanding how plants deploy

RLCKs to cope with pathogen infection will eventually contribute to breeding crops with durable disease resistance.

## Methods

### Plant materials

*N. benthamiana:Cf-4 sobir1/sobir1-like* knock-out plants were used in this study. As *SOBIR1-like* is not functional in *N. benthamiana*, we further refer to these knock-out plants as *N. benthamiana:Cf-4 sobir1*[40].

A highly efficient multiplex editing technique employed to knock-out multiple RLCK-VII subfamily members in *N. benthamiana:Cf-4* has been described previously[84]. To screen for homozygous transformants, genomic DNA from each mutant plant was isolated by using the Phire Tissue Direct PCR Master Mix (Thermo Fisher Scientific), followed by amplifying the sgRNA-targeted regions and subsequent Sanger sequencing of the obtained PCR fragments. Primers used for genotyping can be found in Supplementary Table 2.

### Plant growth conditions

All *N. benthamiana* plants used in this study were cultivated in a climate chamber under 15 h of light at 21 °C and 9 h of darkness at 19 °C, with a relative humidity of ~70%.

### Generation of binary vectors for *Agrobacterium tumefaciens*-mediated transient transformation

The constructs pBIN-KS-35S::*Nb*SOBIR1-eGFP (SOL2911), pBIN-KS-35S::*Nb*SOBIR1$^{D482N}$-eGFP (kinase-dead mutant) (SOL7928), pBIN-KS-35S::*Sl*SOBIR1-eGFP (SOL2774), pBIN-KS-35S::*Sl*SOBIR1$^{D473N}$-eGFP (kinase-dead mutant) (SOL2875), pBIN-KS-35S::*Sl*SOBIR1-like-eGFP (SOL2773), pBIN-KS-35S::*Sl*SOBIR1-like $^{D486N}$-eGFP (kinase-dead mutant) (SOL2876) and pMOG800-Avr4 have been described previously[14]. The codon change, resulting in a Ser/Thr-to-Ala or Tyr-to-Phe amino acid change in the kinase domain of SOBIR1s, was introduced by performing overlap extension PCR using the plasmids pENTR/D-Topo:*Nb*SOBIR1 (SOL4064), pENTR/D-Topo:*Sl*SOBIR1 (SOL2746), and pENTR/D-Topo:*Sl*SOBIR1-like (SOL2745) as templates[14,85]. Phusion Hot Start II DNA Polymerase (Thermo Scientific) was used for the overlap extension PCR and the primers that were used are listed in Supplementary Table 2. The methylated template plasmids remaining in the PCR products were digested by DpnI (NEB), and after transformation to *E. coli* DH5α, the required SOBIR1 mutants carrying individual mutations were selected by Sanger sequencing, and then introduced into pBIN-KS-35S::GWY-eGFP (SOL2095; for C-terminally tagging with eGFP), by using Gateway LR Clonase II (Invitrogen). Thereby, the binary vectors pBIN-KS-35S::*Nb*SOBIR1$^{T512A}$-eGFP (SOL7909), pBIN-KS-35S::*Nb*SOBIR1$^{T515A}$-eGFP (SOL7910), pBIN-KS-35S::*Nb*SOBIR1$^{T516A}$-eGFP (SOL7911), pBIN-KS-35S::*Nb*SOBIR1$^{S517A}$-eGFP (SOL7912), pBIN-KS-35S::*Nb*SOBIR1$^{T522A}$-eGFP (SOL7913), pBIN-KS-35S::*Sl*SOBIR1$^{T503A}$-eGFP (SOL7969), pBIN-KS-35S::*Sl*SOBIR1$^{T506A}$-eGFP (SOL7970), pBIN-KS-35S::*Sl*SOBIR1$^{T507A}$-eGFP (SOL7971), pBIN-KS-35S::*Sl*SOBIR1$^{S508A}$-eGFP (SOL7972), pBIN-KS-35S::*Sl*SOBIR1$^{T513A}$-eGFP (SOL7973), pBIN-KS-35S::*Sl*SOBIR1-like$^{T516A}$-eGFP (SOL7950), pBIN-KS-35S::*Sl*SOBIR1-like$^{T519A}$-eGFP (SOL7951), pBIN-KS-35S::*Sl*SOBIR1-like$^{T520A}$-eGFP (SOL7952), pBIN-KS-35S::*Sl*SOBIR1-like$^{S521A}$-eGFP (SOL7953), pBIN-KS-35S::*Sl*SOBIR1-like$^{T526A}$-eGFP (SOL7954), pBIN-KS-35S::*Nb*SOBIR1$^{Y355F}$-eGFP (SOL7914), pBIN-KS-35S::*Nb*SOBIR1$^{Y426F}$-eGFP (SOL7915), pBIN-KS-35S::*Nb*SOBIR1$^{Y429F}$-eGFP (SOL7916), pBIN-KS-35S::*Nb*SOBIR1$^{Y431F}$-eGFP (SOL7917), pBIN-KS-35S::*Nb*SOBIR1$^{Y469F}$-eGFP (SOL7918), pBIN-KS-35S::*Nb*SOBIR1$^{Y525F}$-eGFP (SOL7919), pBIN-KS-35S::*Nb*SOBIR1$^{Y530F}$-eGFP (SOL7920), pBIN-KS-35S::*Nb*SOBIR1$^{Y543F}$-eGFP (SOL7921), pBIN-KS-35S::*Sl*SOBIR1$^{Y346F}$-eGFP (SOL7974), pBIN-KS-35S::*Sl*SOBIR1$^{Y417F}$-eGFP (SOL7975), pBIN-KS-35S::*Sl*SOBIR1$^{Y420F}$-eGFP (SOL7976), pBIN-KS-35S::*Sl*SOBIR1$^{Y422F}$-eGFP (SOL7977), pBIN-KS-35S::*Sl*SOBIR1$^{Y460F}$-eGFP (SOL7978), pBIN-KS-35S::*Sl*SOBIR1$^{Y521F}$-eGFP (SOL7979), pBIN-KS-35S::*Sl*SOBIR1$^{Y522F}$-eGFP (SOL7980), pBIN-KS-35S::*Sl*SOBIR1$^{Y534F}$-eGFP (SOL7981), pBIN-KS-35S::*Sl*SOBIR1$^{Y588F}$-eGFP

(SOL7982), pBIN-KS-35S::*Sl*SOBIR1-like$^{Y359F}$-eGFP (SOL7942), pBIN-KS-35S::*Sl*SOBIR1-like$^{Y430F}$-eGFP (SOL7943), pBIN-KS-35S::*Sl*SOBIR1-like$^{Y433F}$-eGFP (SOL7944), pBIN-KS-35S::*Sl*SOBIR1-like$^{Y435F}$-eGFP (SOL7945), pBIN-KS-35S::*Sl*SOBIR1-like$^{Y473F}$-eGFP (SOL7946), pBIN-KS-35S::*Sl*SOBIR1-like$^{Y529F}$-eGFP (SOL7947), pBIN-KS-35S::*Sl*SOBIR1-like$^{Y534F}$-eGFP (SOL7948) and pBIN-KS-35S::*Sl*SOBIR1-like$^{Y547F}$-eGFP (SOL7949), for *in planta* expression were obtained.

To clone the selected tomato RLCK homologues for split-luciferase assays, the open reading frames of the encoding genes were amplified from cDNA obtained from leaves of tomato cv. Moneymaker, inoculated with *B. cinerea*[55]. *Sl*SOBIR1 was amplified from pENTR-*Sl*SOBIR1[14], and *GUS* was amplified from pENTR-GUS (Invitrogen). PCR reactions were performed using primers with KpnI and SalI restriction sites for directional cloning into Cluc- and Nluc-vectors[54] (Supplementary Table 2). The Nluc-vector includes an HA-tag for additional immunoblot analyses (Jian-Min Zhou, personal communication). After confirmation by sequencing, the vectors Solyc07G041940-Cluc (Sol7202), Solyc04G082500-Cluc (Sol7204), Solyc11G062400-Cluc (Sol7206), Solyc06G062920-Cluc (Sol7208), Solyc05G025820-Cluc (Sol7210), Solyc01G088690-Cluc (Sol7212), Solyc05G007050-Cluc (Sol7214), Solyc08G077560-Cluc (Sol7216), Solyc01G112220-Cluc (Sol7218), Solyc06G005500-Cluc (Sol7220), SlSOBIR1-Nluc (Sol6766), and GUS-Nluc (Sol6793) were transformed to *A. tumefaciens* strain C58C1, carrying the pCH32 helper plasmid. Generation of the AtBIK1-Cluc (Sol6625) construct was described previously[86].

The bait-TbID fusion proteins were generated using a gateway-compatible 35S-YFP-TbID expression vector based on pEarleyGate101 (pEG101)[87]. The vectors GUS-YFP-TbID (SOL9203) and *Nb*SOBIR1-YFP-TbID (SOL9201) were generated by LR reactions from the gateway entry vectors SOL2685 and SOL4064. The primers that were used are listed in Supplementary Table 2.

For the generation of a complementation construct for RLCK Niben101Scf00012g00012, cDNA of *N. benthamiana* was used to amplify the coding region of the encoding gene with Phusion Hot Start II DNA polymerase, using the primers listed in Supplementary Table 2. The purified PCR product was inserted in pENTR-TOPO and subsequently inserted in the destination vector pBIN-KS-35S::GWY-eGFP (SOL2095), with an LR reaction. The resulting complementation vector was transformed to *A. tumefaciens* C58C1, carrying the pCH32 helper plasmid.

## *A. tumefaciens*-mediated transient transformation

All binary plasmids were transformed into *A. tumefaciens* (further referred to as Agrobacterium) strain C58C1, carrying the helper plasmid pCH32. Agrobacterium strains harbouring the transient expression constructs of RLP23 and RLP42 were received from Lisha Zhang and Thorsten Nürnberger[20,50]. Agrobacterium cells were harvested by centrifugation (3500 × *g*, 15 min) and resuspended in MMAi (1 L of MMAi: 5 g of MS salts, 1.95 g of MES, 20 g of sucrose, and 200 μM acetosyringone) to a final OD$_{600}$ of 0.8. After 1 h incubation at room temperature, cultures were infiltrated into the first fully expanded *N. benthamiana* leaves with a 1-mL disposable syringe[88].

The development of cell death was photographed and quantified by red light imaging using the ChemiDoc (Bio-Rad)[42]. Statistical analysis was performed using one-way ANOVA by GraphPad Prism 9.

## Immunoprecipitation (IP)

The protein accumulation level of SOBIR1 mutants and various tomato RLCKs after their transient expression in leaves of *N. benthamiana* was determined by performing a protein immunoprecipitation assay followed by immunoblotting[89]. For this, leaf samples were harvested at 2 days post infiltration (dpi) and frozen in liquid nitrogen. Subsequently, the leaf samples were ground to a fine powder, after which pre-cooled extraction buffer (150 mM NaCl, 1.0% IGEPAL CA-630 [NP-40], 50 mM Tris, pH 8.0, plus one protease inhibitor tablet per 50 mL extraction buffer) was added to the leaf powder in a 1 g: 2 mL ratio and mixed thoroughly. Samples were then centrifuged at 4 °C for 15 min at 18,000 × *g*, and 2 mL of the supernatant was incubated with 15 μL GFP-trap_A beads at 4 °C for 1 h. Hereafter, the beads were collected by centrifugation at 1000 × *g* for 1 min and washed three times in 1 mL of extraction buffer. After the final wash, SDS loading buffer (200 mM Tris, pH 6.8, 8% SDS, 40% glycerol, 400 mM DTT, 2% Bromophenol blue) was added to the beads and the mixture was boiled at 95 °C for 10 min.

Subsequently, the immunoprecipitated proteins were separated on an SDS-PAGE gel and transferred to a PVDF membrane (Trans-Blot Turbo Transfer Pack, Bio-Rad), using a Trans-Blot Turbo Transfer system (Bio-Rad) (settings: 1.3 A, 25 V, 7 min). The membrane was incubated with TBS-Tween (150 mM NaCl, 20 mM Tris, pH 7.5, 0.1% tween) containing 5% milk powder at room temperature for 1 h or at 4 °C overnight. To detect GFP fusion proteins, the blots were incubated with anti-GFP-HRP (1:5000) (Miltenyi Biotec, 130-091-833). To detect Cluc-tagged proteins, blots were incubated with anti-Cluc (1 μg/mL) (Sigma, L2164) and anti-mouse-HRP (1:10,000) (GE Healthcare).

## Reactive oxygen species (ROS) assay

For ROS burst assays, leaf discs were taken from 4-week-old *N. benthamiana:Cf-4* and *rlck* knock-out plants, while for plants transiently expressing SOBIR1 mutants for the complementation studies, *RLP23* or *RLP42*, leaf discs were collected at 24 h after ago-infiltration. Leaf discs were then floated on 80 μL of sterile water in a 96-well plate overnight and hereafter, the water in each well was replaced carefully by 50 μL of fresh sterile water. After another 1 h of incubation, 50 μL of the reaction solution, containing 100 μM of luminol (L-012, Fujifilm, Japan), 20 μg/mL horseradish peroxidase (Sigma), and the elicitor to be tested (being 0.2 μM Avr4 protein, 0.2 μM flg22, 20 μM chitohexaose, 2 μM nlp20 or 2 μM pg13), was added to each well. Subsequently, the production of luminescence was monitored with a CLARIOstar plate reader (BMG Labtech). The line charts showing the detected values of ROS were created using GraphPad Prism 9.

## MAPK activation assay

Each SOBIR1 mutant was transiently co-expressed with *Avr4* (OD$_{600}$ = 0.8 for each binary vector) in the first fully expanded leaves of *N. benthamiana:Cf-4 sobir1* plants. Leaf samples were harvested at 2 dpi, and leaf samples from the various *N. benthamiana:Cf-4 rlck* knock-out lines, infiltrated with a solution of 5 μM Avr4 protein, were harvested at 15 min after treatment.

Total protein was extracted and subjected to SDS-PAGE, after which an anti-p42/p44-erk antibody (NEB) was employed to detect the activated MAPKs on western blots.

## Recombinant protein expression

To produce the recombinant cytoplasmic kinase domain (KD) of SOBIR1, BAK1 and various RLCKs with a GST or 6 × His tag in *E. coli*, the vectors pET-GST (Addgene No. 42049) and pET-15b were employed, respectively. Both vectors were linearized by PCR amplification with the primer pairs pET-GST_fw/rev and pET-15b_fw/rev (Supplementary Table 2). Meanwhile, the coding sequence of *Nb*SOBIR1 KD, *Sl*SOBIR1 KD, *Sl*SOBIR1-like KD, and several selected RLCKs, as well as all their corresponding kinase-dead mutants, were PCR-amplified from the corresponding pENTR/D-Topo plasmids, using the primers containing the homologous sequence of either pET-GST or pET-15b (Supplementary Table 2). Hereafter, the linearized vector and amplified insert were recombined by using the ClonExpress II One Step Cloning kit (Vazyme, China). After transformation to *E. coli* DH5α, the correct expression constructs were selected by performing colony PCRs and Sanger sequencing.

For recombinant protein expression, the *E. coli* strain BL21 was used. Bacteria harbouring the correct expression construct were cultured at 37 °C overnight in LB liquid medium and subsequently inoculated into fresh LB medium at a ratio of 1:200 (v/v). After culturing at 37 °C for around 3 h, the bacterial population became in an exponential growth phase, with an $OD_{600}$ between 0.6 and 0.8, at which IPTG was added to a final concentration of 0.5 mM, followed by incubating the culture at 22 °C overnight for protein production.

### In vitro phosphorylation assay

In vitro phosphorylation assays were performed as previously described[46]. In brief, cells from 100 μL of the *E. coli* cultures that were started for protein expression were collected by centrifugation and then resuspended in 100 μL of SDS sample buffer (200 mM Tris-HCl, 8% SDS, 40% glycerol, 400 mM DTT, and 0.2% bromophenol blue), followed by boiling for 10 min. Hereafter, the samples were centrifuged for 2 min at maximum speed in an Eppendorf centrifuge, and 8 μL of the supernatant were loaded onto a precast mini-PROTEIN TGX Polyacrylamide Gel (BIORAD). After running for around 100 min at 160 V, the gel was incubated in fixation solution (50% methanol, 10% acetic acid in $H_2O$), overnight. Next, the gel was washed in deionized water for 30 min twice, and the phosphorylated proteins were stained using a Pro-Q Diamond solution (Invitrogen). Subsequently, the staining solution was removed by washing the gel in de-staining solution (20% acetonitrile, 50 mM sodium acetate), and proteins were stained with CBB.

### Phylogenetic analysis of the RLCKs from Arabidopsis, *N. benthamiana*, and tomato

To conduct a phylogenetic analysis of the *At*BIK1 homologues in *N. benthamiana*, tomato, and Arabidopsis, their predicted proteomes were obtained from www.solgenomics.net and www.arabidopsis.org. Hereafter, the three predicted proteomes were independently queried for Pfam domains by using HMMER (v3.1b2; gathering cut-off)[90]. Sequences that contain annotated Pfam domains aside from cytoplasmic kinases (PF00069 or PF07714) were removed, and the sequences of the annotated kinase domains were extracted. Then, we took the domain PF07714 as a lead and removed the sequences that deviated in length from the kinase domain of *At*BIK1. The remaining 1455 kinase domain sequences were aligned using MAFFT (v7.271)[91], the alignment was subsequently trimmed using ClipKIT (v1.3.0; smart-gap)[92] and a neighbour-joining phylogenetic tree was built using QuickTree (with 1000 bootstrap replicates)[93] (Supplementary Fig. 9a). Next, a well-supported (>92% bootstrap support) sub-clade of putative BIK1 homologues, which comprised 123 sequences including *At*BIK1, was extracted from this guide tree. Subsequently, a refined phylogenetic tree was generated with these sequences by using the maximum-likelihood (ML) phylogeny as implemented in IQ-Tree (v2.2.0)[94]. The 123 extracted kinase domains were re-aligned using MAFFT and trimmed as described above, and the ML phylogeny was constructed in IQ-Tree, using automatic amino acid substitution model selection (optimal model: Q.plant with five categories of rate heterogeneity)[95,96]. Branch support for the phylogenetic tree was obtained using ultrafast bootstrap, as well as SH-aLRT, as implemented in IQ-tree[97].

### *Phytophthora palmivora* inoculation assay

Fully expanded leaves were harvested from five-week-old *N. benthamiana* plants and used for inoculation with *Phytophthora palmivora* zoospores. *P. palmivora* maintenance and zoospore production were performed as described before[98]. Inoculations were performed by depositing 10 μL droplets containing 10,000 zoospores each on the abaxial side of the leaves. Leaves were incubated at 21 °C with a 16-h photoperiod in large Petri dishes containing wet paper to maintain moisture. The extent of colonization was monitored at 2 days post inoculation using the necrotic spot areas as a read-out for susceptibility. Necrotic areas were determined by imaging

chlorophyll autofluorescence using a Chemidoc Imaging System (Bio-Rad), and the extent of colonization was quantified using ImageJ (https://imagej.nih.gov/ij).

### Split-luciferase assay

To confirm the proper accumulation of the various tomato RLCKs, fused to the C-terminal part of the luciferase enzyme (Cluc), each binary expression vector was agro-infiltrated with the silencing suppressor P19 into a whole leaf of *N. benthamiana*, at an $OD_{600}$ of 1.0 both for the expression vector and P19. At 3 dpi the leaves were harvested and a protein immunoprecipitation assay, followed by immunoblotting using anti luciferase antibody (Sigma, L2164), was performed as described before[89].

Split-luciferase assays were performed using agro-infiltration of combinations of *Sl*SOBIR1, *At*FLS2 or GUS, fused to the N-terminal part of the luciferase enzyme (Nluc), and selected tomato RLCKs, fused to the C-terminal part of the luciferase enzyme (Cluc) at an $OD_{600}$ of 0.5, as described before[54]. At 3 dpi, leaves were imaged with the abaxial side up using a ChemiDoc (Bio-Rad) device. For this, the leaves were sprayed with luciferin (1 mM luciferin (Biovision, sodium salt 7902-100), dissolved in Milli-Q, supplemented with 1/5000 (v/v) Silwet L-77 (Lehle Seeds, VIS-30)). The leaves were kept in the dark for 5 min to reduce autofluorescence, and luminescence was subsequently detected using the following settings: no illumination, no filter, 2 × 2 binning, and an exposure time of 20 min. A colorimetric image was also made and merged with the luciferase picture.

### Transient expression for TurboID (TbID)-based proximity-dependent labelling and streptavidin-pull down for mass spectrometry analysis

Samples were prepared in triplicate, expressing each TbID-tagged bait protein in 15 *N. benthamiana:Cf-4 sobir1* plants, using one leaf per plant and eventually combining the agro-infiltrated leaves of five plants to generate each individual sample. Two days after agro-infiltration for transient expression of the GUS-YFP-TbID and *Nb*SO-BIR1-YFP-TbID bait constructs in *N. benthamiana sobir1*, the leaves were infiltrated with a solution containing biotin (200 μM, pH = 8, 10 mM MES) and harvested after one hour. For each replicate, the five treated leaves were harvested, combined, wrapped in aluminium foil, and kept in liquid nitrogen or at −80 °C until further use. The frozen samples were ground to a fine powder in liquid nitrogen using a mortar and pestle. Then, the pulverized samples were transferred to pre-cooled 50 mL V-shape tubes, weighed, and resuspended in 2 mL/g of extraction buffer (EB; pH = 8.0, 150 mM NaCl, 1.0% [v/v] IGEPAL®CA-630 (=NP-40), 50 mM Tris, Sigma protease inhibitor cocktail = 1 tablet per 50 mL). For this resuspension, the frozen samples with the EB were vortexed at room temperature and kept on ice once melted. Then, the samples were centrifuged at 18,000 × g for 30 min at 4 °C in a Sigma 4–16 K centrifuge.

To prevent saturation of the streptavidin beads with the free biotin remaining in the samples, 10 mL of the centrifuged protein extracts were desalted using PD MiniTrap PD-10 desalting columns (GE Healthcare). This was done at 4 °C, following the manufacturer's gravity protocol for removing salt.

The desalted protein extracts were then transferred to pre-cooled 15 mL tubes and incubated for one hour at 4 °C with 10 rotations per minute in an SB3 tube rotator (STUART), with 200 μL of Dynabeads™ MyOne™ Streptavidin C1 (washed before use according to the manufacturer's protocol). After incubation, the beads were washed three times with 1 mL of EB, without NP-40, after which they were resuspended in 45 μL of EB buffer.

### Sample preparation for proteomics by mass spectrometry

While still on the beads, the disulphide bonds in the captured proteins were reduced by adding 5 μL of DTT (150 mM) and incubating the

samples at 45 °C for 30 min. The sulfhydryl groups were subsequently alkylated by adding 6 µL of acrylamide (200 mM) and incubating the samples µfor 10 min at room temperature.

The peptides to be measured by mass spectrometry were subsequently released from the streptavidin-coated beads by tryptic digestion. For this, a stock solution of trypsin (0.5 µg/µL of trypsin in 1 mM HCl, pH 3) was diluted 100 times in ABC buffer (ammonium bicarbonate, 50 mM, pH = 8) and 100 µL of the diluted trypsin solution was added to each sample. The samples were then incubated overnight at room temperature, with mild agitation, after which they were acidified to pH = 3 using trifluoroacetic acid and cleaned up using µColumns according to the method published by Wendrich and co-workers[99].

## LC−MS/MS analysis

For the LC−MS/MS analysis, the peptides were separated by reversed phase nano liquid chromatography, using a Thermo nLC1000 system equipped with a home-made C18 nanoLC column and they were measured using an Orbitrap Exploris 480 mass spectrometer. The peptide spectra were searched in Maxquant (version 2.0.3.0) 438, using the Andromeda search engine 439 with label-free quantification (LFQ), against the version Niben1.0.1 of the *N. benthamiana* proteome dataset, including the protein sequence of Cf-4 (O50025), Avr4 (Q00363), LTI6b (AT3G05890), GUS (P05804) and frequently occurring contaminants.

The identified protein groups were then analysed using Perseus (version 1.6.2.3) 358. Reverse and contaminant proteins, and those only identified by matching, were filtered out. Then, protein groups identified in less than three replicate samples were also filtered out. The LFQ values were log2 transformed, and the missing values were assigned assuming a normal distribution. The relative protein quantitation of the samples relative to the controls was calculated applying both-sided Student's t-tests, using a permutation-based adjustment (FDR = 0.05, 250 randomizations, and S0 set to 0.1).

## Reporting summary

Further information on research design is available in the Nature Portfolio Reporting Summary linked to this article.

## Data availability

The authors declare that all data supporting the findings of this study are available within the manuscript and the Supplementary Files or are available from the corresponding authors upon request. Source data are provided with this paper.

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

## Acknowledgements

The authors thank Bert Essenstam from Unifarm for excellent plant care. Laurens Deurhof and Gabriel Lorencini Fiorin are acknowledged for technical assistance. We thank Xiaoqian Shi-Kunne for her help with retrieving the protein sequences of the SOBIR1 homologues from different plant species. We thank Jim Renema, Max Pluis, Agnes Omabour Hagan and Amber van Loosbroek for their help with performing some of the experiments. W.R.H. Huang is supported by the China Scholarship Council (CSC) (201706990001). S.L.V. is supported by the Peruvian Council for Science, Technology and Technological Innovation (CONCYTEC) and its executive unit FONDECYT (119-2017-FONDECYT).

## Author contributions

W.R.H.H., A.M.vd B., and M.H.A.J.J. conceived and designed the experiments. W.R.H.H., C.B., S.L.V., H.L., A.M.vd B., L.Z., Y.W., and E.E. executed the experimental work, data acquisition and analysis. S.B. performed LC-MS/MS and assisted with the analysis of the acquired data. C.K. and F.F. generated *rlck* knock-out lines; M.F.S. performed phylogenetic analysis of BIK1 homologues; J.W. and T.N. provided input; W.R.H.H., M.S., E.E., J.S. and M.H.A.J.J. wrote the manuscript with contributions from all authors.

## Competing interests

The authors declare no competing interests.
