## [Peer Review File · Nature Communications]

Reviewers' Comments:

Reviewer #1:

Remarks to the Author:

The manuscript by Huang and colleagues provides some novel insights into the phospho-regulation of SOBIR1 and BAK1, two central receptor kinases required for activating RLP-triggered immunity in plants. They mainly focus on the functions of *Nicotiana benthamiana* proteins and how SOBIR1 and BAK1 can phosphorylate each other, but show that some Nb-described phosphorylation sites translate to other species as well. They identify two important phosphosites for SOBIR1 function (T522 and Y469) that is important for downstream signal activation. Moreover, they then go one step below the receptor complex and study the function of multiple RLCK-VII subfamilies and their involvement in SOBIR1-dependent (but also independent, e.g. flg22 perception) signaling pathways downstream of ligand perception. Overall, the paper provides some novel insights into SOBIR1/BAK1/RLCK regulation beyond *Arabidopsis*, but lacks sufficient mechanistic details into the interplay of these components. The authors should perform some more specific experiments to improve the story and strengthen their conclusions. Detailed issues are listed below:

Major issues:

- ROS burst assays in general lack a *sobir1* mutant control, since all tested SOBIR1 mutants still show a ROS burst, although reduced: Is there partial restoration of the phenotype or do they not complement at all?
- To make the claim that SOBIR1 transphosphorylates NbBAK1 *in vitro* (and vice versa), they need to include controls. Unrelated receptor kinase cytoplasmic domains (e.g. CERK1) that is not phosphorylated by SOBIR1 should be tested to allow meaningful judgements.
- Since SOBIR1 Y469F mutants are not kinase dead, the authors should test whether the mutation affects BAK1 transphosphorylation or whether the mutation affects interaction with BAK1 *in vivo* (upon ligand perception). This would allow to gain important mechanistic insight into SOBIR1-BAK1 regulation and the relevance of this potential phosphosite.
- Line 319-324: Why did the authors already perform ROS and HR assays in the T1 generation of RLCK mutants? Isn't it expected that multiple genes are rather heterozygous than homozygous making any conclusions difficult? Why not genotyping first and selecting mutants for subsequent generation? Since there is no description what genes are actually mutated in these „T1 lines“, Figure S11-S13 need to be removed or properly explained (what genes were mutated, what was the status of the mutants in individual lines).
- The description of the CRISPR mutants lacks some information about where the mutations are located within the respective gene. This should be added. Also, the numbering of the *rlck* lines is quite confusing and difficult to get from looking at figures. It is hard to understand which genes were actually mutated in individual lines. Please change the numbering (e.g. 734-3-5-5-1) into something more easy to relate to (in that example *rlckVII-6* line 1 or similar). This would strongly increase accessibility of the results.
- All the data analysing the different higher order *rlck* mutants (e.g. Fig 3) would greatly benefit from a transient complementation assays selecting one of the respective members. This would pinpoint whether several members have non-redundant functions that culminate in the same reduced immune output outcome, or whether they are truly redundant.
- A direct correlation between BAK1/SOBIR1 and RLCKs is missing. Do the proteins interact *in vivo*? The authors should test this by CoIP using selected members of the different RLCK clusters. Ideally, they should pick members that were able to complement the *rlck* mutant phenotypes (see comment above). The mere transphosphorylation assay *in vitro* is not sufficient evidence for interaction of these components.
- Overall, the paper lacks some mechanistic insight. How is the critical phosphorylation site T522 in the SOBIR1 activation segment phosphorylated? Is this done by BAK1? Is this residue phosphorylated *in vitro* and/or *in vivo* by BAK1? Is a SOBIR1 T522A mutant affected in BAK1-mediated phosphorylation? Does mutating this site (or, see previous comment, the identified Y469 site) affect interaction between SOBIR1 and BAK1 (upon ligand perception) *in vivo*? These experiments should be performed to add important data to the BAK1-SOBIR1 transphosphorylation model and to strengthen the authors conclusions

Minor points:

- Fig. S4E: Unlike NbSOBIR1, SISOBIR1/SISOBIR1-like seem to require multiple residues in the activation domain to mount full kinase activity. This is not discussed in the manuscript and should be done.

- Line 313-318: I do not understand the reasoning why certain members of RLCK-VII-6 were chosen to be targeted by CRISPR: „Hence, the genes that are up-regulated or remain unchanged.... were selected to be knocked out“. Does that mean that everything that was not DOWNregulated or where any expression was detected was targeted? This needs to be better explained.

- The RLCK phylogenetic tree in Fig. S9 is impossible to grasp. All text is very small and it is hard to get relations between different RLCKs from Arabidopsis with the Nicotiana and tomato ones. The authors should highlight in the tree with bigger text individual RLCKs from Arabidopsis with a known function, e.g. RIPK, BIK1, PBL1.

- Line 392-393: I was confused by the detailed description of the MAPK assay to detect WIPK/SIPK activation upon Avr4 treatment. Are these different MAPKs that are detected compared to results depicted in Fig. 1 or 2? This needs to be clarified

Reviewer #2:

Remarks to the Author:

SOBIR1 and BAK1 act as central signaling sectors that activate the receptor complex through transphosphorylation. In this study, the authors found the transphosphorylation between BAK1 and SOBIR1 leading to the full activation of the SOBIR1/BAK1-containing immune complex in Solanaceae species. SOBIR1 and BAK1 directly trans-phosphorylate receptor-like cytoplasmic kinases members of RLCK-VII-6, -7 and -8 in vitro, which are differentially involved in tomato LRR-RLP Cf-4/ Cladosporium fulvum apoplastic effector Avr4 triggered immune response and hypersensitive response. This study provides novel insights on the signal transduction initiated by membrane-localized immune receptors. I have a few concerns for the author to address, Comments and suggestions:

1. I am wondering whether protein phosphorylation between SOBIR1 and BAK1 occurs in vivo. This should be addressed by additional experimental data or detailed discussion.
2. In *N. benthamiana*, is NbSOBIR1-like acts similar to NbSOBIR1? Same as SISOBIR1-like?
3. In Figure 1e, it is worth to mention that the Avr4-triggered ROS burst on the T516A, S517A, T515A line of NbSOBIR1 is stronger than WT. However, this is not the case for other mutants.
4. Based on the results confirming the importance of T522 of NbSOBIR1 for Avr4/Cf-4 signaling and the presence of transphosphorylation of NbBAK1/NbSOBIR1. It is worth to show or discuss whether the transphosphorylation of NbBAK1/NbSOBIR1 is essential for Avr4/Cf-4 signaling.
5. The figures showing ROS production are difficult to read. It would be better to add a bar graph for all the ROS data, or to make the different lines more distinguishable (Figure 1e, S2e, S2i, S8a, S8b, S11, S12 etc).
6. Western blot images need to be labeled with protein size.
7. Line 334-336, it would be better to name the independent homozygous lines with a more concise number. Furthermore, if rlck-vii-8 769-4-4-20-3 and 769-4-4-20-8 lines were from the 769-4 in Figure S11e, no information on 769-4-4 in Figure S11e. It would be better to mention why choosing these independent homozygous lines for further analyses.
8. In Figure S14, knockout lines showed clear dwarfed phenotype when compared to WT. Could this plant growth phenotype influence the ROS burst.

REBUTTAL TO REVIEWER COMMENTS

Reviewer #1 (Remarks to the Author):

The manuscript by Huang and colleagues provides some novel insights into the phospho-regulation of SOBIR1 and BAK1, two central receptor kinases required for activating RLP-triggered immunity in plants. They mainly focus on the functions of *Nicotiana benthamiana* proteins and how SOBIR1 and BAK1 can phosphorylate each other, but show that some Nb-described phosphorylation sites translate to other species as well. They identify two important phosphosites for SOBIR1 function (T522 and Y469) that is important for downstream signal activation. Moreover, they then go one step below the receptor complex and study the function of multiple RLCK-VII subfamilies and their involvement in SOBIR1-dependent (but also independent, e.g. flg22 perception) signaling pathways downstream of ligand perception. Overall, the paper provides some novel insights into SOBIR1/BAK1/RLCK regulation beyond *Arabidopsis*, but lacks sufficient mechanistic details into the interplay of these components. The authors should perform some more specific experiments to improve the story and strengthen their conclusions. Detailed issues are listed below:

Major issues:

- ROS burst assays in general lack a *sobir1* mutant control, since all tested SOBIR1 mutants still show a ROS burst, although reduced: Is there partial restoration of the phenotype or do they not complement at all?

ANSWER:

All SOBIR1 mutants were tested in the sobir1 knock-out mutant. The kinase-dead and the T522A SOBIR1 mutants do not complement the ROS burst, which is completely abolished in the sobir1 knock-out, whereas the Y469F mutant does restore this ROS burst.

Please see Figures 1 and 2;

Figure 1b-e, Complementation with NbSOBIR1 T522A fails to restore Avr4/Cf-4-triggered HR (b and c), MAPK activation (d), and ROS burst (e) in N. benthamiana:Cf-4 sobir1 knock-out plants.

Fig 2e: Transient expression of NbSOBIR1 Y469F restores the Avr4/Cf-4-triggered ROS accumulation in N. benthamiana:Cf-4 sobir1 knock-out plants.

The kinase-dead mutant and T522A still trigger some ROS, as all SOBIR1 mutants were tested through agro-infiltration. For this complementation study we first transiently expressed the individual SOBIR1 mutants in the sobir1 knock-out plants and measured ROS at 2 dpi. The transient expression by agro-infiltration always results in some background ROS.

- To make the claim that SOBIR1 transphosphorylates NbBAK1 in vitro (and vice versa), they need to include controls. Unrelated receptor kinase cytoplasmic domains (e.g. CERK1) that is not phosphorylated by SOBIR1 should be tested to allow meaningful judgements.

ANSWER:

The unrelated receptor kinase CERK1 does not recruit SOBIR1 in planta upon its activation, so the cytoplasmic kinase domain of CERK1 will never have the chance to be phosphorylated by SOBIR1 in vivo, as they will never be in close proximity in real life. It could well be that in vitro phosphorylation between the cytoplasmic kinase domains of CERK1 and SOBIR1 will take place, as they are combined in one tube and one kinase can phosphorylate another kinase as its substrate. With the experiments depicted in Figure 1g and 1h, we just aimed to show that trans-phosphorylation between the cytoplasmic kinase domains of

BAK1 and SOBIR1 is actually possible. We do not claim here that this is specific, but just show that it can occur and that BAK1 and SOBIR1 can function as each other's substrate.

- Since SOBIR Y469F mutants are not kinase dead, the authors should test whether the mutation affects BAK1 transphosphorylation or whether the mutation affects interaction with BAK1 in vivo (upon ligand perception). This would allow to gain important mechanistic insight into SOBIR1-BAK1 regulation and the relevance of this potential phosphosite.

ANSWER:

*The SOBIR1 Y469F mutant still has intrinsic kinase activity, so we anticipate that it will be able to interact with BAK1 and phosphorylate the cytoplasmic kinase domain of BAK1. As the SOBIR1 Y469F mutant restores the ROS burst (Figure 2e), for which BAK1 phosphorylation, followed by RBOHB phosphorylation through some phosphorylated RLCK(s), is required, this proves that the SOBIR1 Y469F mutant is still able to interact with BAK1 and can still trans-phosphorylate the cytoplasmic kinase domain of BAK1. We suggest that the SOBIR1 Y469F mutant might not be able to recruit some essential downstream signalling partners (such as particular RLCKs) anymore. This suggestion is supported by our observation that co-expression of NbSOBIR1 Y469F with Avr4 failed to restore the Avr4/Cf-4-triggered MAPK activation in *N. benthamiana*:Cf-4 sobir1 knock-out plants (Figure 2d). In another project, we will be identifying the downstream signalling proteins that can still interact with the kinase domain of the SOBIR1 Y469F mutant, when compared with wild-type and kinase-dead SOBIR1.*

- Line 319-324: Why did the authors already perform ROS and HR assays in the T1 generation of RLCK mutants? Isn't it expected that multiple genes are rather heterozygous than homozygous making any conclusions difficult? Why not genotyping first and selecting mutants for subsequent generation? Since there is no description what genes are actually mutated in these „T1 lines“, Figure S11-S13 need to be removed or properly explained (what genes were mutated, what was the status of the mutants in individual lines).

ANSWER:

*The T1 lines of rick-vii-6 were already genotyped by the team of Dr. Johannes Stuttmann, as described in Stuttmann et al. Highly efficient multiplex editing: one-shot generation of 8x *Nicotiana benthamiana* and 12x *Arabidopsis* mutants. *Plant J* 106, 8-22, doi:10.1111/tpj.15197 (2021). They genotyped the T1 generation of rick-vii-6 and showed that almost all the targeted genes were already mutated, making this multiplex editing approach very efficient. For that reason, we already performed a ROS screen with the transformants of the T1 generation.*

- The description of the CRISPR mutants lacks some information about where the mutations are located within the respective gene. This should be added. Also, the numbering of the rick lines is quite confusing and difficult to get from looking at figures. It is hard to understand which genes were actually mutated in individual lines. Please change the numbering (e.g. 734-3-5-5-1) into something more easy to relate to (in that example rickVII-6 line 1 or similar). This would strongly increase accessibility of the results.

ANSWER:

We have now changed the numbering of the rick lines into: rick-vii-6 #1, #2 and #3; rick-vii-7 #1 and #2; rick-vii-8 #1 and #2, throughout the text and in the figures.

- All the data analysing the different higher order rick mutants (e.g. Fig 3) would greatly benefit from a transient complementation assays selecting one of the respective members. This would pinpoint whether

several members have non-redundant functions that culminate in the same reduced immune output outcome, or whether they are truly redundant.

ANSWER:

We have now added the result of a complementation assay with the RLCK-VII-8 family member Niben101Scf00012g00012.1, which has been shown to interact with the kinase domain of SOBIR1 in split luciferase and proximity-dependent labelling assays (see below). Please see the added text on pages 14 and 20 and Figure S20.

- A direct correlation between BAK1/SOBIR1 and RLCKs is missing. Do the proteins interact in vivo? The authors should test this by CoIP using selected members of the different RLCK clusters. Ideally, they should pick members that were able to complement the rick mutant phenotypes (see comment above). The mere transphosphorylation assay in vitro is not sufficient evidence for interaction of these components.

ANSWER:

As the interactions between RLCKs and the kinase domain of BAK1 and/or SOBIR1 are weak and transient, a regular co-IP might not be sufficient to show interaction. Therefore, we performed split-luciferase assays to show in vivo interactions between RLCKs and SOBIR1. This revealed that several tomato RLCKs do interact with the kinase domain of SISOBIR1. We have now added this information to our manuscript, on pages 13, 19, 23 and Figure S18.

We also performed TurboID-based proximity-dependent labelling by biotinylation, in combination with mass spectrometry, using NbSOBIR1 as a TurboID-fused bait, in N. benthamiana. This has revealed that various N. benthamiana RLCKs are biotinylated, suggesting that these RLCKs interact with the kinase domain of NbSOBIR1. We have added these data on pages 14, 20, 23/24/25 and Figure S19.

These data collectively show that various RLCKs and the kinase domain of SOBIR1 interact in planta.

- Overall, the paper lacks some mechanistic insight. How is the critical phosphorylation site T522 in the SOBIR activation segment phosphorylated? Is this done by BAK1? Is this residue phosphorylated in vitro and/or in vivo by BAK1? Is a SOBIR1 T522A mutant affected in BAK1-mediated phosphorylation? Does mutating this site (or, see previous comment, the identified Y469 site) affect interaction between SOBIR1 and BAK1 (upon ligand perception) in vivo? These experiments should be performed to add important data to the BAK1-SOBIR1 transphosphorylation model and to strengthen the authors conclusions.

ANSWER:

*We have already published that trans-phosphorylation takes place between BAK1 and SOBIR1; see Burgh, A. M. v. d., Postma, J., Robatzek, S. & Joosten, M. H. A. J. Kinase activity of SOBIR1 and BAK1 is required for immune signalling. *Molecular Plant Pathology* **20**, 410-422, doi:10.1111/mpp.12767 (2019), and we also refer to this publication in the text. Furthermore, Wei et al. (2022) (Wei, X. et al. Structural analysis of receptor-like kinase SOBIR1 revealed mechanisms that regulate its phosphorylation-dependent activation. *Plant Communications*, doi:10.1016/j.xplc.2022.100301 (2022), we refer to this paper in the text) have also shown that SOBIR1 is phosphorylated by BAK1, as they describe the following: "Biochemical studies revealed that SOBIR1 is transphosphorylated by BAK1 following its autophosphorylation via an intermolecular mechanism, and the phosphorylation of Thr529 in the activation segment and the b3-aC loop are critical for SOBIR1 phosphorylation. Further functional analysis confirmed the importance of Thr529 and the b3-aC loop for the SOBIR1-induced cell death response in *Nicotiana benthamiana*. Taken together, these findings provide a structural basis for the regulatory mechanism of SOBIR1 and reveal the important elements and phosphorylation events in the special stepwise activation of SOBIR1-KD, the first such processes found in regulatory LRR-RLKs."*

Indeed, Thr522 was found to be important for the auto-phosphorylation of SOBIR1. In Arabidopsis, Wei et al. (2022) show that BAK1 phosphorylates SOBIR1 on this residue, which is Thr529 in AtSOBIR1. Please see the table taken from Wei et al. (2022) below.

Supplemental Table 2. Phosphorylated sites identified by LC-MS/MS

Proteins	Identified Sites
SOBIR1-KD	Thr390, Ser394, Ser406, Thr519, Thr522, Thr523, Ser524, Thr529, Ser566, Thr573, Thr587, Ser592
PP2Co-dephosphorylated SOBIR1	no phosphorylation sites
autophosphorylated dSOBIR1-KD	Thr390, Ser394, Ser406, Thr529
BAK1-phosphorylated dSOBIR1-KD	Thr390, Ser394, Ser406, Thr410, Thr519, Thr522, Thr523, Ser524, Thr529, Ser592
SOBIR1-phosphorylated	Thr324, Thr446, Thr449, Thr450, Thr455, Ser595,
BAK1-CD ^{Δ49N}	Ser602, Thr603, Ser604

Thr522A is in fact a kinase-dead mutant and we have already shown earlier that a SOBIR1 kinase-dead mutant still interacts with BAK1 (van der Burgh, A. M. v. d., Postma, J., Robatzek, S. & Joosten, M. H. A. J. Kinase activity of SOBIR1 and BAK1 is required for immune signalling. *Molecular Plant Pathology* **20**, 410-422, doi:10.1111/mpp.12767 (2019), DOI : 10.1111/mpp.12767), and we refer to this publication in the text. So, we anticipate that mutating this site will not affect the interaction between SOBIR1 and BAK1, and that therefore the Thr522A mutant will still interact with the cytoplasmic kinase domain of SOBIR1.

Concerning Y469, Fig 2e from our manuscript shows that complementation by using WT or the Y469F mutant results in a similar Avr4/Cf-4-induced ROS accumulation, for which BAK1 phosphorylation is required, so the interaction between the cytoplasmic kinase domains of SOBIR1 and BAK1 will not be affected by mutating Y469.

We have now added some of this additional information to the discussion section of our manuscript, see lines 508-520.

Minor points:

- Fig. S4E: Unlike NbSOBIR1, SISOBIR1/SISOBIR1-like seem to require multiple residues in the activation domain to mount full kinase activity. THIS is not discussed in the manuscript and should be done.

ANSWER:

We do not see that in Fig S4E. As shown in Fig. S2a, identical phosphorylatable amino acids are present in the activation domain of SISOBIR1, SISOBIR1-like and NbSOBIR1. In all cases, five amino acids can be phosphorylated in the activation domain and mutating them has similar effects on the ROS burst (See Figs 1e and S2e and S2i). Fig S4E shows that the kinase domain of wild-type SIBAK1 directly phosphorylates the kinase domain of the kinase-dead D473N mutant of SISOBIR1.

- Line 313-318: I do not understand the reasoning why certain members of RLCK-VII-6 were chosen to be targeted by CRISPR: „Hence, the genes that are up-regulated or remain unchanged.... were selected to be knocked out“. Does that mean that everything that was not DOWNregulated or where any expression was detected was targeted? This needs to be better explained.

ANSWER:

We have now explained that better. Please see lines 302-310.

- The RLCK phylogenetic tree in Fig. S9 is impossible to grasp. All text is very small and it is hard to get relations between different RLCKs from Arabidopsis with the Nicotiana and tomato ones. The authors should

highlight in the tree with bigger text individual RLCKs from Arabidopsis with a known function, e.g. RIPK, BIK1, PBL1.

ANSWER:

We have now indicated these RLCKs in red and mention this in the legend.

- Line 392-393: I was confused by the detailed description of the MAPK assay to detect WIPK/SIPK activation upon Avr4 treatment. Are these different MAPKs that are detected compared to results depicted in Fig. 1 or 2? This needs to be clarified.

ANSWER:

The MAPKs that are detected in Figure 1d, 2d and 4a are the same MAPKs in all cases. We have now removed the mentioning of WIPK/SIPK from the text (lines 384-387) and from Figure 4.

Reviewer #2 (Remarks to the Author):

SOBIR1 and BAK1 act as central signaling sectors that activate the receptor complex through transphosphorylation. In this study, the authors found the transphosphorylation between BAK1 and SOBIR1 leading to the full activation of the SOBIR1/BAK1-containing immune complex in Solanaceae species. SOBIR1 and BAK1 directly trans-phosphorylate receptor-like cytoplasmic kinases members of RLCK-VII-6, -7 and -8 in vitro, which are differentially involved in tomato LRR-RLP Cf-4/ Cladosporium fulvum apoplastic effector Avr4 triggered immune response and hypersensitive response. This study provides novel insights on the signal transduction initiated by membrane-localized immune receptors. I have a few concerns for the author to address,

Comments and suggestions:

1. I am wondering whether protein phosphorylation between SOBIR1 and BAK1 occurs in vivo. This should be addressed by additional experimental data or detailed discussion.

ANSWER:

We have already shown earlier that Arabidopsis thaliana (At) SOBIR1, which constitutively activates immune responses when overexpressed in planta, is highly phosphorylated. Moreover, in addition to the kinase activity of SOBIR1 itself, kinase-active BAK1 is essential for AtSOBIR1-induced constitutive immunity and for the phosphorylation of AtSOBIR1.

Furthermore, the defence response triggered by the tomato LRR-RLP Cf-4 on perception of Avr4 from the extracellular pathogenic fungus Fulvia fulva is dependent on kinase-active BAK1. We argue that, in addition to the trans-autophosphorylation of SOBIR1, it is likely that SOBIR1 and BAK1 transphosphorylate, and thereby activate the receptor complex. This has been described in van der Burgh et al., 2019 (Burgh, A. M. v. d., Postma, J., Robatzek, S. & Joosten, M. H. A. J. Kinase activity of SOBIR1 and BAK1 is required for immune signalling. Molecular Plant Pathology 20, 410-422, doi:10.1111/mpp.12767 (2019)), to which we also refer in the text.

In addition, Wei et al. (2022) have also shown that SOBIR1 is phosphorylated by BAK1, as they describe the following: "Biochemical studies revealed that SOBIR1 is transphosphorylated by BAK1 following its autophosphorylation via an intermolecular mechanism, and the phosphorylation of Thr529 in the activation segment and the b3-aC loop are critical for SOBIR1 phosphorylation. Further functional analysis confirmed the importance of Thr529 and the b3-aC loop for the SOBIR1-induced cell death response in Nicotiana benthamiana. Taken together, these findings provide a structural basis for the regulatory mechanism of SOBIR1 and reveal the important elements and phosphorylation events in the special stepwise activation of SOBIR1-KD, the first such processes found in regulatory LRR-RLKs." This is

described in: Wei, X. et al. Structural analysis of receptor-like kinase SOBIR1 revealed mechanisms that regulate its phosphorylation-dependent activation. *Plant Communications*, doi:10.1016/j.xplc.2022.100301 (2022), and we refer to this paper in the text. We have now added some of this additional information to the discussion section of our manuscript, see lines 508-520.

2. In *N. benthamiana*, is NbSOBIR1-like acts similar to NbSOBIR1? Same as SISOBIR1-like?

ANSWER:

NbSOBIR1-like is a pseudogene, and is not expressed. This is what we published earlier in PNAS (Liebrand et al., 2013) (Liebrand, T. W. H. et al. Receptor-like kinase SOBIR1/EVR interacts with receptor-like proteins in plant immunity against fungal infection. Proceedings of the National Academy of Sciences of the United States of America 110, doi:10.1073/pnas.1313401110 (2013):

*"The Avr4-triggered HR was also strongly compromised when NbSOBIR1 was targeted. When NbSOBIR1-like was targeted, the HR was affected to a much lesser extent. Quantitative RT-PCRs (qRT-PCRs) revealed that expression of NbSOBIR1 was strongly reduced upon inoculation with TRV:NbSOBIR1/NbSOBIR1-like or TRV:NbSOBIR1, compared with inoculation with TRV:β-glucuronidase (GUS). Interestingly, we did not detect transcripts of NbSOBIR1-like in TRV:GUS-inoculated or TRV:NbSOBIR1/NbSOBIR1-like-inoculated plants, suggesting that NbSOBIR1-like is not expressed or is at a very low level. We therefore reasoned that the slight reduction of the Avr4-triggered HR upon inoculation of *N. benthamiana*:Cf-4 with TRV:NbSOBIR1-like (Fig. 2) could be attributed to cross-silencing of NbSOBIR1 by the TRV: NbSOBIR1-like construct. Indeed, qRT-PCR confirmed that NbSOBIR1 expression levels were ~30%reduced upon inoculation with TRV:NbSOBIR1-like. Together these results indicate that NbSOBIR1 is the RLK that is required for the Cf-4- mediated HR in *N. benthamiana*."*

*These observations are supported by the fact that when knocking out only SOBIR1 in *N. benthamiana*:Cf-4 plants, with the SOBIR1-like gene still being there, the Avr4/Cf-4-triggered HR and ROS are abolished in these plants, indicating that SOBIR1-like is not functional (Huang, W. R. H., Schol, C., Villanueva, S. L., Heidstra, R. & Joosten, M. H. A. J. Knocking out SOBIR1 in *Nicotiana benthamiana* abolishes functionality of transgenic receptor-like protein Cf-4. *Plant Physiology*, doi:10.1093/plphys/kiaa047 (2021)).*

SISOBIR1 and SISOBIR1-like are both expressed in tomato, and seem to act redundantly (Liebrand et al., 2013).

3. In Figure 1e, it is worth to mention that the Avr4-triggered ROS burst on the T516A, S517A, T515A line of NbSOBIR1 is stronger than WT. However, this is not the case for other mutants.

ANSWER:

We do not mention this, as these differences are not reproducible. We show a representative picture of an experiment that has been performed several times, and the ROS burst triggered by T516A, S517A and T515A of NbSOBIR1 is not always stronger than the ROS triggered by WT SOBIR1. It is also not the point that we want to make here. What is essential here, is that the NbSOBIR1 T522A mutant does not complement.

4. Based on the results confirming the importance of T522 of NbSOBIR1 for Avr4/Cf-4 signalling and the presence of transphosphorylation of NbBAK1/NbSOBIR1. It is worth to show or discuss whether the transphosphorylation of NbBAK1/NbSOBIR1 is essential for Avr4/Cf-4 signalling.

ANSWER:

Our observations indicate that indeed the trans-phosphorylation that is expected to take place between BAK1 and SOBIR1 is essential for Avr4/Cf-4 signalling. As published by Liebrand et al., 2013 in PNAS, we observed that the Avr4-triggered HR is strongly compromised when NbSOBIR1 is targeted by virus-induced gene silencing (VIGS).

Furthermore, our research has shown that after elicitation with matching effector ligands Avr4 and Avr9, BAK1 associates with Cf-4 and Cf-9, respectively. BAK1 is required for the effector-triggered hypersensitive response and resistance of tomato against *F. fulva* (Postma et al. Avr4 promotes Cf-4 receptor-like protein association with the BAK1/SERK3 receptor like kinase to initiate receptor endocytosis and plant immunity. *New Phytologist*, doi:10.1111/nph.13802 (2016).

In addition, the defence response triggered by the tomato LRR-RLP Cf-4 upon perception of Avr4 is dependent on kinase-active BAK1. Transient co-expression of Avr4 in *N. benthamiana*:Cf-4 with the dominant negative BAK1 mutants AtBAK1C408Y or AtBAK1D416N results in a reduced Cf-4/Avr4 HR, when compared with co-expression with wild-type AtBAK1 or GUS.

We have now added some of this additional information to the discussion section of our manuscript, see lines 508-520.

5. The figures showing ROS production are difficult to read. It would be better to add a bar graph for all the ROS data, or to make the different lines more distinguishable (Figure 1e, S2e, S2i, S8a, S8b, S11, S12 etc).

ANSWER:

For figures S12 and S13, we have now added the total photon count. For Figures 1e, S2e, S2i, S8a, and S8b, we now only show WT, kinase-dead (KD) and the mutant of interest, which makes the graphs more clear.

6. Western blot images need to be labeled with protein size.

ANSWER:

We have now labelled the protein molecular weights in all figures showing western blots.

7. Line 334-336, it would be better to name the independent homozygous lines with a more concise number. Furthermore, if rlck-vii-8 769-4-4-20-3 and 769-4-4-20-8 lines were from the 769-4 in Figure S11e, no information on 769-4-4 in Figure S11e. It would be better to mention why choosing these independent homozygous lines for further analyses.

ANSWER:

We have now renamed the various RLCK knock-out lines of *N. benthamiana*, and we chose these lines because these were homozygous.

8. In Figure S14, knockout lines showed clear dwarfed phenotype when compared to WT. Could this plant growth phenotype influence the ROS burst.

ANSWER:

The morphological phenotypes of *N. benthamiana*:Cf-4 and the three independent rlck-vii-6, two independent rlck-vii-7, and two independent rlck-vii-8 knock-out lines indeed suggest that there is some slight dwarfing occurring in all these knock-out lines. This might have some general effect on the ROS burst, but the different RLCK knock-out plants show very different ROS burst patterns, so it is not likely that there is a specific effect of the dwarfing on the ROS burst.

Reviewers' Comments:

Reviewer #1:

Remarks to the Author:

I have reviewed the paper before. The authors addressed the majority of raised concerns sufficiently and improved the story, but a few issues remain. Details are listed below:

- The authors now provide Spli-LUC assays to show interaction between S1RLCKs and SISOBIR1 (Fig. S18). Yet, the data is difficult to interpret, as positive luciferase emission is evident on almost all depicted leaves. The authors should perform quantification of the result for statistical analysis of independent replicates.
- The authors now show complementation for AVR4-triggered ROS production of the r1ck-vii-8 #1 knockout mutant by expression of Nb00012, but the experiment lacks a WT (*N. benthamiana*:Cf-4) plant, as e.g. shown in Fig. 3. It is difficult to interpret the degree of complementation without the use of the appropriate „WT“ background for comparison. I agree that the biphasic ROS (as e.g. shown in Fig. 3C) is not observable upon Nb00012 expression, but total photon counts might still be comparable.

Reviewer #2:

Remarks to the Author:

I am satisfied with the revision, and have only two minor concerns:

1, The RLCK VII-8 knockout mutant showed slightly increased ethylene, the authors can discuss the potentials for this phenotype.

2, Mutation of Tyr469 in NbSOBIR1 reduce the HR and MAPK triggered by Avr4/Cf4, but not influence ROS. Please add discussion on how this site influences function of NbSOBIR1 as well as the immune output.

REBUTTAL TO REVIEWER COMMENTS

Reviewer #1 (Remarks to the Author):

I have reviewed the paper before. The authors addressed the majority of raised concerns sufficiently and improved the story, but a few issues remain. Details are listed below:

- The authors now provide Spli-LUC assays to show interaction between S1RLCKs and SISOBIR1 (Fig. S18). Yet, the data is difficult to interpret, as positive luciferase emission is evident on almost all depicted leave. The authors should perform quantification of the result for statistical analysis of independent replicates.

ANSWER:

We did quantify the luciferase emission levels from independent replicates. However, these data ended up in the raw dataset that we submitted but was not visible to the reviewers. We have now added these data (see the figure below) as additional panels (panels **b** to **I**) to Figure S18.

- The authors now show complementation for AVR4-triggered ROS production of the r1ck-vii-8 #1 knockout mutant by expression of Nb00012, but the experiment lacks a WT (*N. benthamiana*:Cf-4) plant, as e.g. shown in Fig. 3. It is difficult to interpret the degree of complementation without the use of the appropriate „WT“ background for comparison. I agree that the biphasic ROS (as e.g. shown in Fig. 3C) is not observable upon Nb00012 expression, but total photon counts might still be comparable.

ANSWER:

We have now added the ROS profile of the “wild type (WT)” (*N. benthamiana*:Cf-4), and associated total photon count, to respectively panels **a** and **b** of Figure S20. It is clear that the total photon count for WT is much higher, indicating that only partial

complementation takes place of the Avr4-triggered ROS production in the *rlck-vii-8* #1 knockout mutant upon expression of Nb00012.

Reviewer #2 (Remarks to the Author):

I am satisfied with the revision, and have only two minor concerns:

1, The RLCK VII-8 knockout mutant showed slightly increased ethylene, the authors can discuss the potentials for this phenotype.

ANSWER:

We have now added the following text to the manuscript (lines 396-399):

“It is worth noting that the *rlck-vii-8* knock-out lines show a slightly increased ethylene production, implying that members of RLCK-VII-8 might be inhibitors of ethylene signalling, triggered by Avr4/Cf-4 (Figure 4f).”

2, Mutation of Tyr469 in NbSOBIR1 reduce the HR and MAPK triggered by Avr4/Cf4, but not influence ROS. Please add discussion on how this site influences function of NbSOBIR1 as well as the immune output.

ANSWER:

In fact, this observation is already addressed in the discussion. We have now made this more clear by adding our finding that the Tyr469Phe mutant of SOBIR1 is still kinase-active.

Lines 571-575 now read:

“Therefore, this important Tyr residue in the kinase domain of SOBIR1 might not regulate plant immune responses by phosphorylation, **as the Tyr469Phe mutant of SOBIR1 is still kinase-active**, but by interacting with specific downstream signalling partners, such as members of RLCK-VII-7 that are required for Avr4/Cf-4-induced HR (Figure 4h).”

So, possibly the Tyr469Phe mutant of SOBIR1 has not lost its interaction with RBOHB, resulting in its phosphorylation and subsequent stimulation of ROS production, but the mutant is not able to interact anymore with an RLCK that is responsible for MAPK activation. This all remains speculation and in the future we aim to study this by performing Turbo-ID-based proximity-dependent labelling studies with the Tyr469Phe mutant, and will compare its interactors with the SOBIR1 wild-type protein.